# TAZ couples Hippo/Wnt signalling and insulin sensitivity through *Irs1* expression

Jun-Ha Hwang[1], A Rum Kim[1], Kyung Min Kim[1], Jung Il Park[1], Ho Taek Oh[1], Sung A Moon[1], Mi Ran Byun[1], Hana Jeong[2], Hyo Kyung Kim[2], Michael B. Yaffe[3], Eun Sook Hwang[2] & Jeong-Ho Hong[1]

Insulin regulates blood glucose levels by binding its receptor and stimulating downstream proteins through the insulin receptor substrate (IRS). Impaired insulin signalling leads to metabolic syndrome, but the regulation of this process is not well understood. Here, we describe a novel insulin signalling regulatory pathway involving TAZ. TAZ upregulates IRS1 and stimulates Akt- and Glut4-mediated glucose uptake in muscle cells. Muscle-specific TAZ-knockout mice shows significantly decreased *Irs1* expression and insulin sensitivity. Furthermore, TAZ is required for Wnt signalling-induced *Irs1* expression, as observed by decreased *Irs1* expression and insulin sensitivity in muscle-specific APC- and TAZ-double-knockout mice. TAZ physically interacts with c-Jun and Tead4 to induce *Irs1* transcription. Finally, statin administration decreases TAZ, IRS1 level and insulin sensitivity. However, in myoblasts, the statin-mediated decrease in insulin sensitivity is counteracted by the expression of a constitutively active TAZ mutant. These results suggest that TAZ is a novel insulin signalling activator that increases insulin sensitivity and couples Hippo/Wnt signalling and insulin sensitivity.

[1] Department of Life Sciences, School of Life Sciences and Biotechnology, Korea University, Seoul 02841, South Korea. [2] College of Pharmacy, Ewha Womans University, Seoul 03760, South Korea. [3] David H. Koch Institute for Integrative Cancer Research, Department of Biology, Massachusetts Institute of Technology, Cambridge, MA 02139, USA. Correspondence and requests for materials should be addressed to E.S.H. (email: eshwang@ewha.ac.kr) or to J.-H.H. (email: jh_hong@korea.ac.kr)

Insulin resistance is a condition wherein cells do not respond appropriately to insulin, further characterized by a risk of developing metabolic syndrome such as cardiovascular disease and type 2 diabetes. Skeletal muscles constitute a major organ for insulin-stimulated glucose uptake and disposal under normal conditions. Under physiological conditions, insulin activates glucose uptake by stimulating the canonical IRS-PI3K-AKT pathway, which stimulates glucose transporter (GLUT) 4 translocation to the membrane for glucose uptake[1,2].

Transcriptional coactivator with PDZ-binding motif (TAZ) and Yes-associated protein (YAP) regulate cell proliferation, differentiation, and stem cell maintenance in response to diverse signalling pathways, including the Hippo, Wnt, GPCR, and mechanotransduction pathways[3–6]. TAZ/YAP are phosphorylated by the LATS kinases, resulting in proteolytic degradation and cytosolic localization by binding to 14-3-3 proteins. Inactivation of Hippo signalling stabilises TAZ/YAP, facilitating TAZ/YAP nuclear localization and interaction with several transcription factors, including members of the transcriptional enhancer factor TEF family (TEADs). TAZ/YAP regulate the transcription of diverse target genes, including connective tissue growth factor (CTGF) and cysteine-rich angiogenic inducer 61 (CYR61)[7–15].

Recently, it was reported that TAZ/YAP activity is regulated by metabolic and nutrient-sensing pathways, suggesting that metabolic status is another factor regulating TAZ/YAP activity[16]. Further, TAZ/YAP activity is regulated by the mevalonate pathway, which is responsible for producing biochemical precursors of isoprenoids. The product of the mevalonate pathway, geranylgeranyl-pyrophosphate, facilitates the membrane localization of Rho protein, which stimulates TAZ/YAP through an unclear mechanism[17,18]. Increased glucose metabolism and reprogramming toward aerobic glycolysis stimulate TAZ/YAP transcriptional activity[19]. AMPK activation by energy stress leads to YAP phosphorylation and inhibits YAP-mediated transcriptional activation through TEADs[20,21]. It has also been reported that AMPK phosphorylates and stabilises AMOTL1, which contributes to YAP inhibition[22]. These reports suggest that TAZ/YAP function as mediators of metabolic signalling.

In this study, we report that TAZ facilitated glucose uptake and increased insulin sensitivity in response to Hippo/Wnt signalling, suggesting that TAZ is a novel regulator of the insulin signalling pathway. Furthermore, the insulin sensitivity-lowering effect of statins, a class of lipid-lowering medications, is regulated via TAZ.

## Results

**TAZ stimulates insulin signalling and increases insulin sensitivity.** To understand the metabolic function of TAZ, muscle-specific TAZ-knockout (mKO) mice were generated using muscle creatine kinase-Cre mice (Supplementary Fig. 1). Insulin-dependent glucose utilization, which primarily occurs in tissues such as muscle, is a process that requires activation of the insulin receptor (IR) and the sequential stimulation of IRS1/2, Akt kinase, and substrates such as ribosomal S6 kinase (S6K) and Akt substrate of 160 kDa (AS160)[23]. To study the role of TAZ in insulin signalling, wild-type (WT) and mKO mice were infused with insulin, and components of the insulin signalling regulatory pathway were analysed. As shown in Fig. 1a, b, mKO mice exhibited lower Akt activity than WT mice did, and this result was confirmed by the decreased phosphorylation of S6K and AS160. In addition, IRS1, but not IRS2, was significantly downregulated in mKO mice, without changes in IR protein level. Similar results were observed in mouse embryonic fibroblasts (MEFs) and C2C12 myoblasts (Fig. 1c, d). IRS1 level and Akt activity were decreased in muscle tissue in mKO mice, but not in other tissues (Supplementary Fig. 2). Next, we analysed *Irs1*

transcription and observed that it was decreased in TAZ-mKO mice, TAZ-knockout (KO) MEFs, and TAZ-knockdown C2C12 myoblasts (Fig. 1e). To further probe the role of TAZ in IRS1 expression, TAZ was reintroduced into TAZ-KO MEFs, which restored Akt activity, IRS1 level (Supplementary Fig. 3a), and *Irs1* transcription (Supplementary Fig. 3b). TAZ was also reintroduced into TAZ knockdown C2C12 myotubes. The introduction restored IRS1 level (Supplementary Fig. 4a), *Irs1* transcription (Supplementary Fig. 4b), and 2-deoxyglucose uptake (Supplementary Fig. 4c). Thus, these results suggest that TAZ stimulates *Irs1* transcription.

To study the role of TAZ in glucose homoeostasis, 2-deoxyglucose uptake was quantified in C2C12 myoblasts in vitro. As shown in Fig. 2a, glucose uptake decreased significantly in TAZ-knockdown cells. Akt induces Glut4 plasma membrane expression to facilitate glucose uptake in skeletal muscles[24]. As illustrated in Fig. 2b, membrane Glut4 levels decreased in TAZ-knockdown cells, suggesting that TAZ stimulates glucose uptake through Glut4 translocation to the plasma membrane. In addition, TAZ KO MEFs exhibited decreased AKT activity and insulin-mediated glucose uptake; however, AKT activity and glucose uptake were rescued upon reintroduction of TAZ or IRS1 into these cells (Supplement Fig. 3c and 3d). We then investigated the role of TAZ in insulin sensitivity in vivo. After glucose infusion, blood glucose disposal decreased in mKO mice (Fig. 2c). Moreover, insulin-mediated reduction in blood glucose levels was also significantly decreased in mKO mice (Fig. 2d). Impaired glucose tolerance and insulin sensitivity in mKO mice did not result from decreased serum insulin levels, as these were slightly increased in mKO mice (Fig. 2e). In addition, this impaired sensitivity could not be attributed to differences in muscle mass because the total muscle size and mass were similar to those of WT and mKO mice, as indicated via β-dystroglycan staining and muscle weight (Supplementary Fig. 5a, 5b and 5c). Muscle fibre type composition was also analysed in accordance with marker gene expression of each fibre types (Type I for slow twitch fibre, Type II for fast twitch fibre). There were no significant differences in muscle fibre type composition between WT and mKO muscle (Supplementary Fig. 5d).

Further, we assessed a metabolic phenotype of WT and mKO mice. On administering standard laboratory chow, no metabolic disorder was detected, except for a difference in insulin sensitivity. There were no phenotypic differences in liver and fat tissue. However, on a high-fat diet, we observed several phenotypic differences. TAZ mKO mice showed decreased insulin sensitivity and effective glucose disposal (Supplementary Fig. 6a and 6b). Increased body weight of mKO mice was observed with increased adipocyte size (Supplementary Fig. 6c and 6d). Interestingly, mKO mice developed a severe fatty liver phenotype, evident from an increased amount of fat droplets in liver sections (Supplementary Fig. 6e). Thus, these results suggest that TAZ is an important regulator of insulin signalling in muscles, thereby influencing whole-body metabolism.

**TAZ stimulates *Irs1* transcription with c-Jun and Tead4.** To identify the *Irs1* regulatory elements at which TAZ binds, chromatin immunoprecipitation (ChIP) sequencing was performed using FLAG-TAZ (F-TAZ)-expressing C2C12 cells. Among the TAZ-binding DNA elements, the primary target element (TAZ binding element #1; TBE1) was selected by considering the peak score of ChIP sequencing and the chromatin modification status of the genomic region flanking *Irs1*, including ChIP sequencing data related to H3Kme1, H3Kme3, and H3K27ac, obtained from the NCBI Sequence Read Archive (Fig. 3a). The transcriptionally

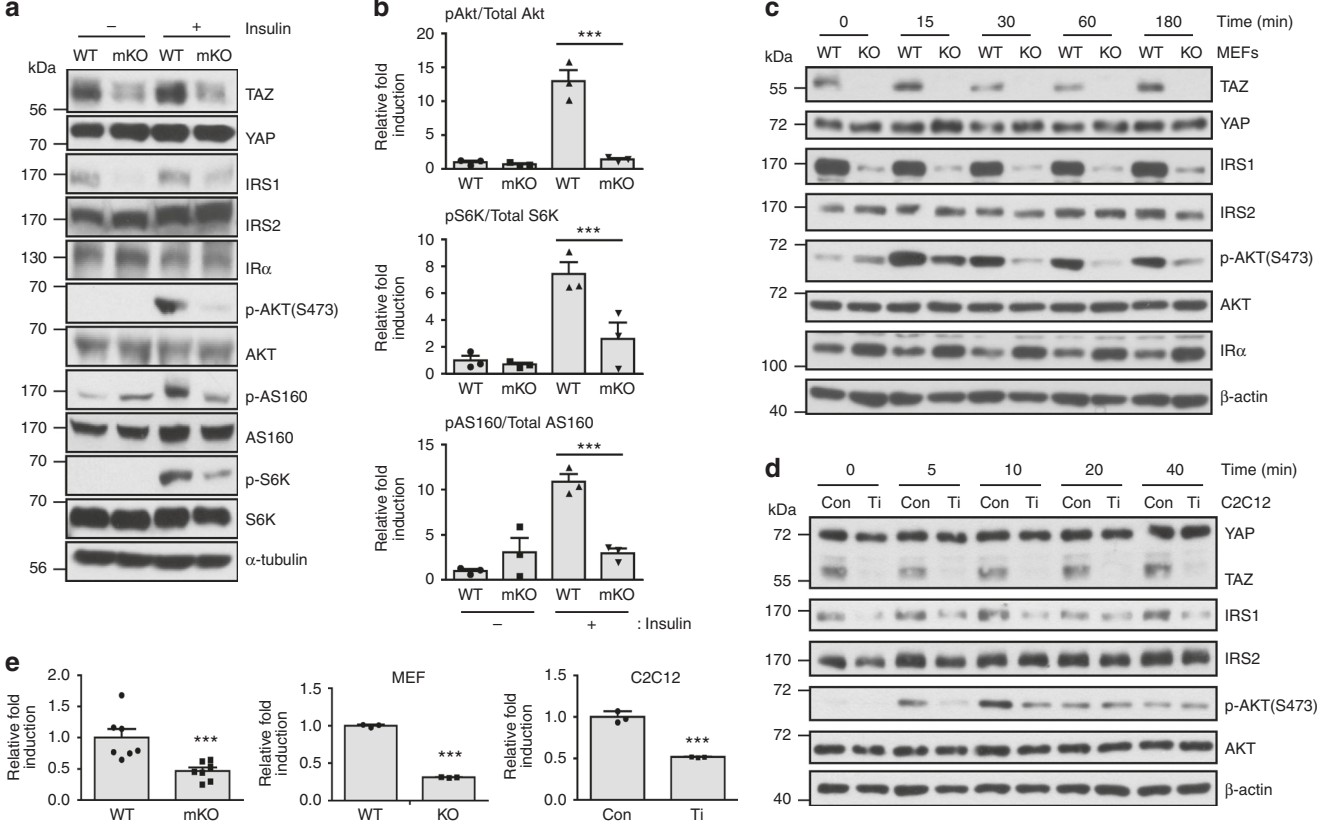

**Fig. 1** TAZ stimulates *Irs1* expression and insulin signalling. **a** Insulin was intraperitoneally injected into wild-type (WT) and muscle-specific TAZ-knockout (mKO) mice. After 15 min, muscle lysates were analysed by immunoblotting. Eight- to ten-week-old mice were used for experiment. **b** The three independent experiments shown in **a** were assessed, and the ratio of phosphorylated to total AKT was analysed, as well as the protein levels of ribosomal protein S6 kinase and AS160. **c** Serum-starved WT and TAZ-knockout (KO) mouse embryonic fibroblasts (MEFs) were treated with 1 nM insulin. Cell lysates were analysed by immunoblotting at the indicated time points. β-Actin was used as a loading control. **d** Serum-starved control (Con) and TAZ-knockdown (Ti) C2C12 myoblasts were treated with 1 nM insulin. Cell lysates prepared at the indicated time points were analysed by immunoblot analysis. β-Actin was used as a loading control. **e** *Irs1* transcription was analysed by quantitative reverse transcription (qRT)-PCR in skeletal muscle from WT and mKO mice (n = 7, left), WT and TAZ-KO MEFs (n = 3, middle), Con and Ti C2C12 cells (n = 3, right). For skeletal muscle data, 8–10-week-old mice were used. Data are presented as mean ± SD for panels **b** and **e** (middle and right), and mean ± SEM for panel **e** (left). Statistical analysis was performed using a Student's t-test. *p < 0.05; ***p < 0.005

active region overlapped with TBE1, suggesting that TAZ is recruited to transcriptionally active *Irs1 cis*-regulatory elements. As shown in Fig. 3b, TAZ binding to TBE1 was validated via ChIP-PCR. Furthermore, using the PROMO and JASPAR programs, we analysed the TBE1 sequence to identify predicted transcription factor binding sites. C/EBPα, c-Jun, YY1, and TEAD were identified as putative transcription factor binding factors (Supplementary Fig. 7a). siRNAs were used to verify whether these transcription factors regulate *Irs1* expression. As shown in Supplementary Fig. 7b and 7c, c-Jun and Tead4 siRNAs effectively downregulated *Irs1*. However, depletion of C/EBPα or YY1 did not affect *Irs1* expression. In addition, we observed that c-Jun depleted cells decrease TAZ inducing IRS1 level (Supplementary Fig. 8a) and *Irs1* transcription (Supplementary Fig. 8b). These results suggest that c-Jun and Tead4 regulate *Irs1* expression. Similarly, it has been reported that c-Jun and Tead4 cooperatively stimulate TAZ/YAP target genes[25,26]. Thus, we used siRNAs to deplete both c-Jun and Tead4 and observed that their depletion significantly downregulated *Irs1* (Fig. 3c, d). Finally, in muscles, we observed that TAZ is localized on the TBE1 of *Irs1* in the absence of the insulin signal, which was not observed in mKO mice. After 15 min of insulin treatment, TAZ binding increased slightly on the TBE1 (Supplementary Figure 9). These results suggest that TAZ is important for basal *Irs1* expression. Next, to

study the transcriptional regulatory activity of TBE1, approximately 500 bp of the DNA elements surrounding TBE1 were cloned and inserted into a luciferase reporter plasmid. As shown in Fig. 3e, reporter gene activity was induced in the presence of TAZ and either c-Jun or Tead4. Additive stimulation was observed in the presence of TAZ and both c-Jun and Tead4. Furthermore, when the reporter plasmids were introduced into TAZ-knockdown cells, decreased reporter gene activity was observed (Fig. 3f). Thus, the results suggest that TAZ is a transcriptional co-stimulator of c-Jun- and Tead4-mediated *Irs1* transcription.

A ChIP assay was performed to verify whether c-Jun is directly recruited to TBE1. As shown in Fig. 4a, TBE1 was occupied by c-Jun. c-Jun contains a Pro-Pro-X-Tyr motif, which can interact with the TAZ WW domain[27]. Thus, we determined whether TAZ physically interacts with c-Jun. As shown in Fig. 4b, ectopically expressed TAZ and c-Jun interacted with each other. As shown in Fig. 4c, TAZ and c-Jun interacted endogenously. Finally, the interaction of TAZ and c-Jun (Supplementary Fig. 10a), or TAZ and Tead4 (Supplementary Fig. 10b) was observed in muscles. We also generated a luciferase reporter plasmid containing a mutation in the c-Jun-binding site and observed that the mutation significantly decreased reporter gene activity, even in the presence of TAZ and c-Jun (Fig. 4d). Additional data revealed

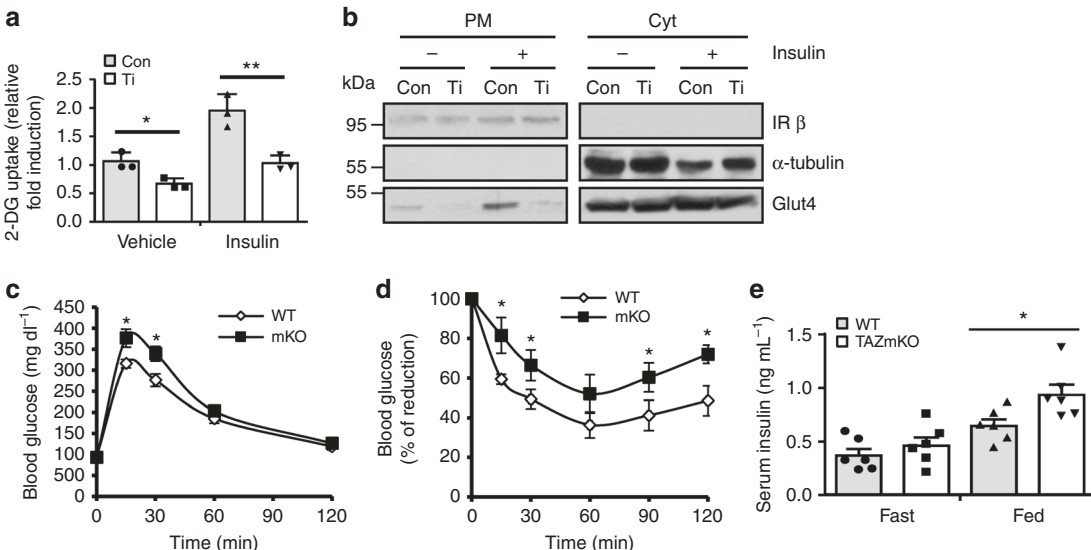

**Fig. 2** TAZ stimulates glucose uptake and increases insulin sensitivity. **a** Glucose uptake was analysed in control (Con) and TAZ-knockdown (Ti) C2C12 myotubes ($n = 3$). **b** Con and Ti C2C12 myotubes were treated with 100 nM insulin. After 30 min, the plasma membrane (PM) and cytosol (Cyt) were fractionated and analysed by immunoblotting using Glut4 antibodies. The insulin receptor β-subunit and α-tubulin were used as PM and Cyt loading controls, respectively. **c** For the glucose tolerance test, WT and mKO mice were fasted for 16 h, and D-glucose was injected intraperitoneally. Blood glucose was measured at the indicated time points ($n = 6$, left). **d** For the insulin tolerance test, WT and mKO mice were fasted for 4 h, and insulin was administered intraperitoneally. Blood glucose levels were measured at the indicated time points ($n = 7$, right). **e** WT and mKO mice were fasted for 16 h or fed ad libitum, and serum insulin concentrations were analysed ($n = 6$). For **c–e**, 8–10-week-old mice were used. Data are presented as mean ± SD for panel **a**, and mean ± SEM for panels **c–e**. Statistical analysis was performed using a Student's $t$-test. *$p < 0.05$; **$p < 0.01$

that the TAZ WW domain is important for interacting with c-Jun. As shown in Fig. 4e, TAZ constructs lacking the WW domain (TAZΔWW and TAZ164-395) did not interact with c-Jun, but WW domain-containing constructs (TAZwt and TAZ124-395) did interact with c-Jun. In addition, c-Jun-mediated reporter gene activity was increased by the TAZ constructs containing the WW domain (Fig. 4f). Thus, the results suggest that the interaction of TAZ and c-Jun is important for *Irs1* transcription.

**TAZ stimulates Wnt signalling-induced *Irs1* expression.** Insulin signalling cross-talks with the Wnt signal[28] and altered Wnt signalling components impair glucose metabolism and diabetes. A genome-wide association study reported that polymorphisms in Wnt5B, Wnt co-receptors Lrp5/6, and Wnt signalling transcriptional cofactor, TCF7L2, are associated with an increased risk of metabolic syndromes[29–32]. In addition, Wnt signalling stimulates *Irs1* expression and insulin signalling[28,33]. Given that TAZ is known as a mediator of Wnt signalling[34–36], we investigated whether TAZ plays a role in Wnt signalling-induced *Irs1* expression. WT and TAZ-KO MEFs were cultivated with Wnt3a, and 24 h later, a significant induction of *Irs1* expression in WT MEFs was observed. This did not occur in TAZ-KO MEFs, suggesting that TAZ is required for Wnt3a-mediated IRS1 induction (Fig. 5a, b). In addition, Wnt3a-induced *Irs1* expression was not observed in TAZ-knockdown C2C12 myotubes (Fig. 5c, d). To study the role of TAZ in the Wnt signalling-induced *Irs1* expression in vivo, mice carrying floxed alleles for adenomatous polyposis coli (APC), a negative regulator of Wnt signalling, were crossed with TAZmKO mice to produce muscle-specific TAZ and APC double-knockout mice (DbmKO). As shown in Fig. 5e, IRS1 level in the gastrocnemius muscles was higher in APCmKO mice than in WT mice, but it was significantly decreased in DbmKO mice. Decreased *Irs1* expression was also observed in DbmKO mice (Fig. 5f). Thus, these results suggest that TAZ is required for

Wnt signalling-induced *Irs1* expression. APCmKO mice exhibited significant blood glucose clearance activity, but this activity was decreased in DbmKO mice (Fig. 5g). In addition, APCmKO mice exhibited significantly increased insulin sensitivity, whereas DbmKO mice displayed decreased sensitivity (Fig. 5h). The results suggest that Wnt signalling-induced insulin sensitivity is regulated by TAZ.

**Statin decreases *Irs1* expression and insulin sensitivity through TAZ downregulation.** TAZ activity is regulated by the mevalonate pathway, through which statins (HMG-CoA reductase inhibitors) inhibit TAZ nuclear localization and decrease TAZ level[17]. Thus, we investigated whether statin-derived TAZ inhibition alters insulin sensitivity. For the experiment, mice were administered simvastatin, and TAZ and IRS1 level was analysed. As shown in Fig. 6a, level of TAZ and of IRS1 was decreased in simvastatin-treated mice, and the drug decreased *Irs1* transcription in these animals (Fig. 6b). Next, we investigated the role of simvastatin in insulin sensitivity in vivo. Blood glucose clearance in simvastatin-treated mice was significantly lower than that in control mice (Fig. 6c). In addition, insulin-mediated reduction of blood glucose levels was inhibited in simvastatin-treated mice (Fig. 6d). Statin-mediated decreases in TAZ and IRS1 level were also observed with C2C12 myotubes. As illustrated in Fig. 6e, f, simvastatin-treated C2C12 myotubes exhibited decreased TAZ and IRS1 levels and *Irs1* transcription. Similar results were observed following treatment with the cerivastatin (Fig. 6g, h).

To further study the effect of simvastatin on insulin signalling, control and simvastatin-treated mice were infused with insulin, and components of the insulin signalling regulatory pathway were analysed. As shown in Fig. 7a, b, simvastatin-treated mice exhibited lower Akt activity than control mice did, and this result was confirmed by the decreased phosphorylation of AS160 and S6K. These results provide a reason for impaired glucose tolerance and insulin sensitivity in simvastatin-treated mice.

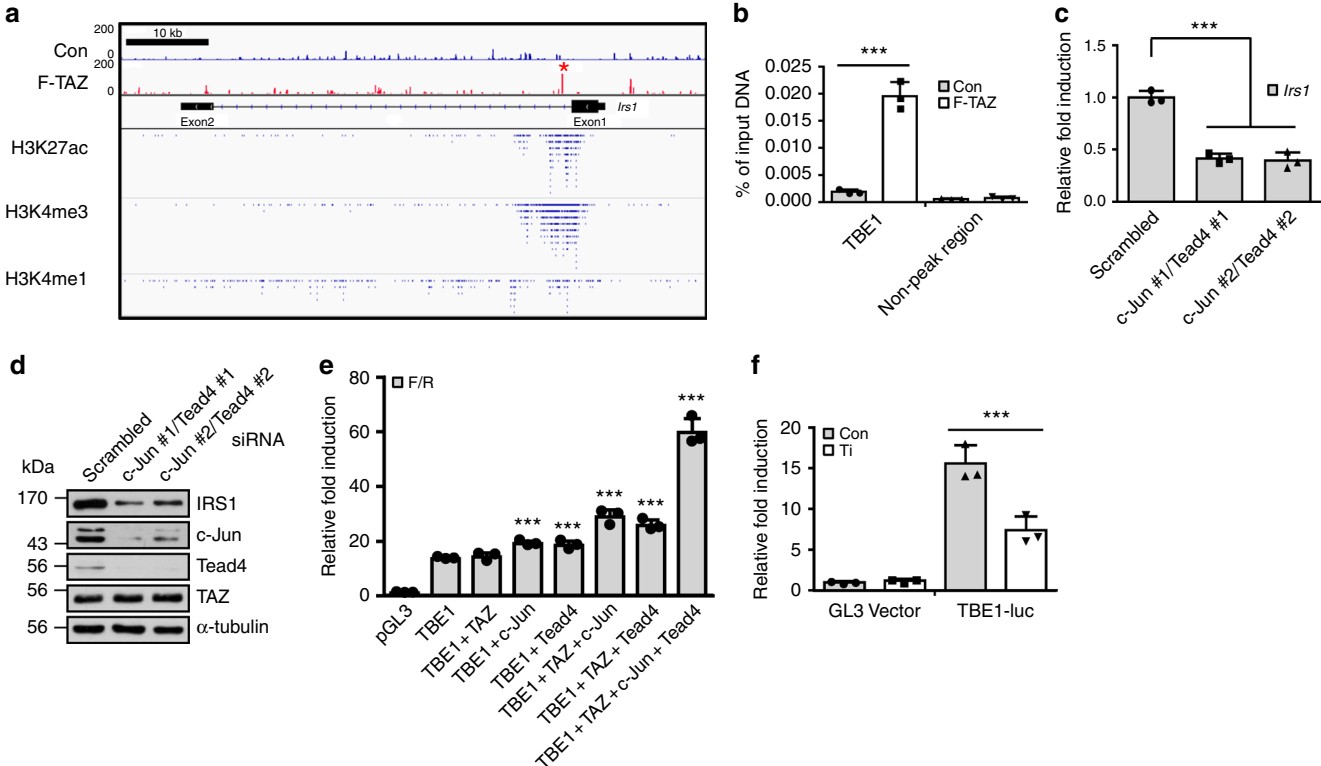

**Fig. 3** TAZ stimulates *Irs1* transcription with c-Jun and Tead4. **a** Chromatin immunoprecipitation (ChIP) sequencing analysis was assessed in FLAG-tagged TAZ (F-TAZ)-overexpressing or control C2C12 myoblasts. Sites of F-TAZ binding were visualised in the mouse genome. The primary target site for TAZ-binding (TAZ-binding element 1, TBE1) is marked with an asterisk. The histone modification status (H3K27ac, H3K4 me3, and H3K4 me1) was obtained from raw ChIP sequencing read data downloaded from the NCBI Sequence Read Archive (SRP009088). **b** Chromatin immunoprecipitation (ChIP)-quantitative PCR (qPCR) was performed to validate the TAZ-binding element 1 (TBE1) site. ChIP was performed in control and F-TAZ-overexpressing C2C12 cells. ChIPed DNA was analysed by qPCR with primer sets spanning the TBE1 site ($n = 3$). **c** Scrambled or two different siRNAs for both c-Jun and Tead4 were co-transfected into C2C12 cells. Transfected cells were analysed by qRT-PCR for *Irs1* transcripts 48 h after transfection ($n = 3$). **d** Depleting c-Jun and Tead4 decreased IRS1 level. C2C12 cells were treated with two different siRNAs for c-Jun and Tead4 and lysates were analysed by immunoblotting. **e** TAZ and c-Jun stimulate a TBE1-containing luciferase reporter gene. Approximately 500 bp of intronic elements surrounding TBE1 were cloned into the pGL3-basic luciferase vector (TBE1-luc), and the completed plasmid was transfected into 293T cells together with c-Jun-, Tead4-, and/or TAZ-expressing plasmids. Luciferase activity was analysed 24 h after transfection. The pGL3-basic vector was used as a control ($n = 3$). **f** TAZ stimulated TBE1-driven gene transcription. Control and TAZ-knockdown 293T cells (Ti) were transfected with pGL3-basic or TBE1-luc, and a luciferase assay was performed 24 h after transfection ($n = 3$). Data are presented as mean ± SD. Statistical analysis was performed using a Student's *t*-test. ***$p < 0.005$

Next, to determine whether inhibiting simvastatin-mediated downregulation of *Irs1* expression is regulated by TAZ, WT TAZ or proteolytic degradation-resistant TAZ (TAZ4SA) were stably expressed in C2C12 myotubes, and *Irs1* expression was analysed. As shown in Fig. 7c, d, overexpressed WT TAZ was degraded in the presence of simvastatin, which decreased *Irs1* expression. However, in TAZ4SA-overexpressing cells, no reduction in *Irs1* expression or transcription was observed. In addition, glucose uptake was not decreased in the TAZ4SA-overexpressing cells after simvastatin treatment (Fig. 7e). The results suggest that statin-mediated reduction in insulin sensitivity is due to decreased IRS1 level and AKT activity and that TAZ depletion is responsible for the decreased insulin sensitivity.

## Discussion

This study reports a hitherto unknown function of TAZ: stimulation of insulin sensitivity through *Irs1* expression. This increases AKT activity in response to insulin, which stimulates glucose uptake through inducing membrane localization of Glut4 (Figs. 1 and 2). The results reveal that TAZ stimulates insulin signalling through IRS1. Previously, it was shown that TAZ and YAP activity is regulated by metabolic and nutrient-sensing

pathways[16]. Increased glucose metabolism stimulates TAZ/YAP transcriptional activity[19]. AMPK activation by energy stress leads to YAP inhibition[20–22]. Thus, our results suggest that TAZ is actively involved in glucose uptake via increasing a metabolic signalling component such as IRS1, as well as a sensor of metabolic signals in response to energy stress. Notably, we observed that TAZ-mediated insulin sensitivity is inhibited by statins (Figs. 6 and 7). Statins are widely used to treat hyperlipidaemia and to prevent cardiovascular diseases[37,38]. However, meta-analyses of clinical trials revealed a variable incidence of new-onset diabetes mellitus[39–42], and several studies report that the diabetogenic action of statins is linked to decreased insulin sensitivity[43,44]. Our study showed that statin-derived alterations in insulin sensitivity are mediated by TAZ (Figs. 6 and 7). Thus, decreased TAZ level may be a potential cause for statin-driven diabetogenic action and provide a mechanistic rationale for finding novel statin derivatives to decrease blood cholesterol levels without side effects.

However, prenylation of small GTPases including Rho and Rab4 facilitates their anchoring to the plasma membrane, which is important for GLUT4 translocation to the plasma membrane for glucose uptake. Statins inhibit HMG-CoA reductase to decrease mevalonate levels and ultimately down-regulate the prenylation of

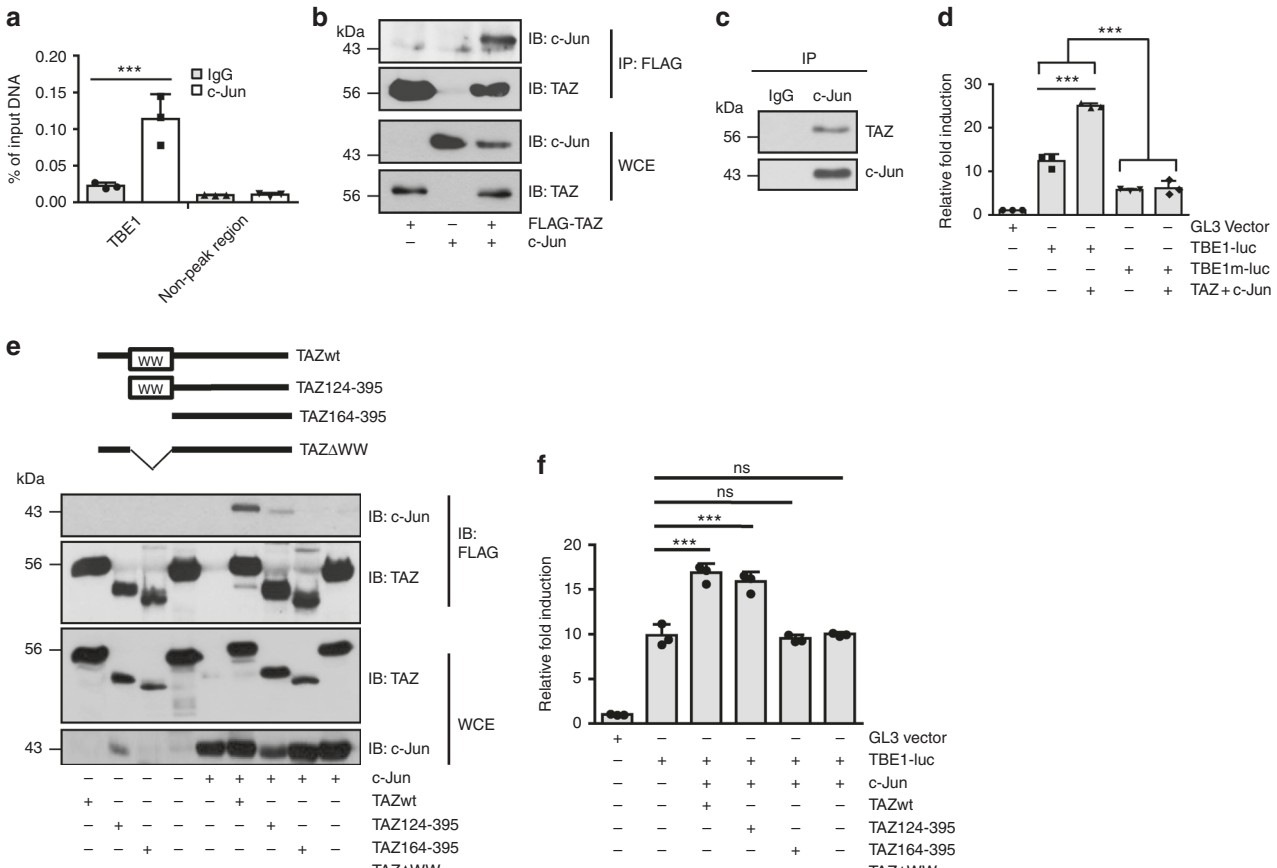

**Fig. 4** TAZ interacts with c-Jun for *Irs1* transcription. **a** Mouse gastrocnemius muscle genomic DNA was sheared by sonication, and chromatin fragments were immunoprecipitated with c-Jun antibodies. ChIPed DNA was analysed by qRT-PCR with primer sets flanking TBE1 ($n = 3$). Eight- to ten-week-old mice were used. **b** TAZ interacted with c-Jun. 293T cells were co-transfected with F-TAZ- and c-Jun-expressing plasmids. After immunoprecipitating with anti-FLAG antibodies, the eluted samples were analysed by immunoblotting with TAZ and c-Jun antibodies. **c** Endogenous c-Jun interacted with TAZ. C2C12 cell lysates were immunoprecipitated with IgG or c-Jun antibodies, and immune complexes were detected using TAZ and c-Jun antibodies. **d** Mutating c-Jun-binding sites in TBE1-luc decreased TAZ- and c-Jun-mediated reporter activity. The mutant c-Jun-binding site was generated by site-directed mutagenesis. Luciferase reporter plasmids harbouring wild-type (TBE-luc) or mutant c-Jun-binding sites (TBE1m-luc) were transfected into 293T cells together with c-Jun- and TAZ-expressing plasmids. Luciferase reporter gene activity was analysed 24 h after transfection ($n = 3$). **e** Wild-type (amino acids 1–395), N-terminus-deleted TAZ (amino acids 124–395 and 164–395) and WW domain-truncated TAZ expression plasmids (ΔWW) were transfected into 293T cells together with a c-Jun expression plasmid. Cell lysates were immunoprecipitated 24 h after transfection with FLAG antibodies, and the eluted samples were analysed by immunoblotting using TAZ and c-Jun antibodies. **f** The transcriptional activity of the TAZ deletion mutant was assessed using a TBE1-luc reporter plasmid. 293T cells were co-transfected with the TAZ deletion mutant together with a c-Jun expression plasmid as described in **e**. A *Renilla* luciferase plasmid was used for transfection normalisation. Cells were lysed, and luciferase activity was measured 24 h after transfection ($n = 3$). Data are presented as mean ± SD. Statistical analysis was performed using Student's *t*-test. ns not significant; \*\*\*$p < 0.005$

Rho and Rab4 GTPase. Thus, it is also possible that other pathways regulated by statin play a certain role in stain-induced insulin resistance. In this study, we did not investigate the effect of TAZ in Rab4 prenylation, which may yield a robust conclusion regarding the role of TAZ in statin-induced insulin resistance.

In addition to insulin sensitivity, our results suggest a novel signalling axis for organ size control. IRS1 is an important factor for organ size control in response to insulin/IGF signalling. *Drosophila* mutants for Chico, which encodes an IRS protein homologue, exhibit reduced cell number and size[45]. IRS1 KO mice exhibit slowed embryonal and postnatal growth with resistance to the glucose-lowering effects of insulin[46]. As TAZ is an effector molecule of Hippo signalling, which regulates organ size[4,47,48], our results suggest that a noble TAZ-IRS1 axis regulates organ size in response to Hippo signalling. Cells communicate signals for growth and proliferation with each other, and our study also provided evidence for signal crosstalk. We observed that Wnt signalling-induced *Irs1* expression is mediated

by TAZ (Fig. 5). Wnt signalling stimulates cell growth by activating the tuberous sclerosis complex-target of rapamycin (TOR) pathway, which are downstream effectors of insulin signalling[49]. Wnt activates TOR kinase to stimulate S6K via inhibiting glycogen synthase kinase 3 (GSK3). Because TAZ is activated and stabilised by Wnt signalling[34–36], our results suggest that TAZ acts as a signalling mediator for a crosstalk with Wnt and the insulin pathway to regulate cell growth and proliferation.

Recently, multiple members of the mitogen-activated protein kinase kinase kinase kinase (MAP4K) were identified as parallel Hippo signalling kinases, which phosphorylates LATS kinase[50–52]. Among them, MAP4K4 has been shown to be a negative regulator of insulin signalling; the silencing of MAP4K4 in adipocytes elevated the expression of GLUT4[53]. In addition, the silencing in human skeletal muscle prevented tumour necrosis factor-α-induced insulin resistance[54]. Interestingly, common polymorphisms of the *MAP4K4* locus are associated with type 2 diabetes and insulin resistance[55]. These

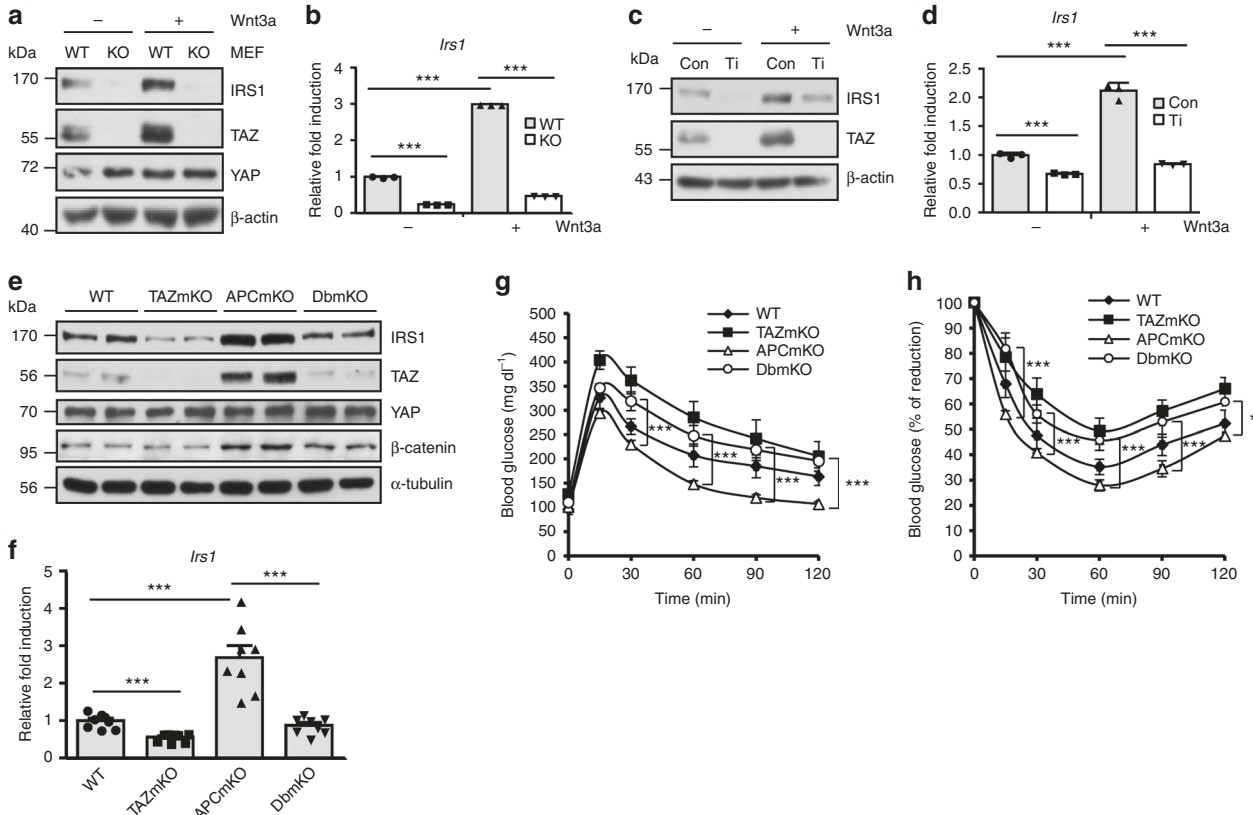

**Fig. 5** TAZ stimulates Wnt signalling-induced *Irs1* transcription. **a** Wild-type (WT) and TAZ-knockout (KO) mouse embryonic fibroblasts (MEF) were treated with Wnt3a-conditioned or control medium from L cells, for 24 h. Cell lysates were analysed by immunoblotting. **b** *Irs1* expression of cells described in **a** was analysed by qRT-PCR (*n* = 3). **c** Control (Con) and TAZ-knockdown (Ti) C2C12 myotubes were treated with control or Wnt3a-conditioned medium for 24 h. Level of IRS1 and TAZ was then analysed by immunoblotting. **d** Total RNAs from cells described in **c** were prepared, and *Irs1* expression was analysed by qRT-PCR (*n* = 3). **e** Gastrocnemius muscles from WT, muscle-specific TAZ-knockout (TAZmKO), muscle-specific adenomatous polyposis coli (APC)-KO (APCmKO), and muscle-specific APC and TAZ double-KO (DbmKO) mice were analysed by immunoblotting, using the indicated antibodies. For β-catenin data, non-phospho (active) β-catenin antibody was used. **f** *Irs1* transcription was analysed in the gastrocnemius muscles of WT, TAZmKO, APCmKO, and DbmKO mice by qRT-PCR (*n* = 8). **g** Glucose tolerance and **h** insulin tolerance were assessed in WT, TAZ-mKO, APC-mKO, and DbmKO mice (*n* = 6). For **e**–**h**, 8–10-week-old mice were used. Data are presented as mean ± SD for panels **b** and **d**, and as mean ± SEM for panels **f**–**h**. Statistical analysis was performed using a Student's *t*-test. *$p < 0.05$; ***$p < 0.005$

results suggest the important role of Hippo signalling regulators in insulin signalling.

Taken together, our data demonstrate that TAZ regulates *Irs1* transcription and stimulates insulin sensitivity, thus identifying TAZ as a metabolic regulator and revealing a link between the Hippo/Wnt signalling pathway and insulin sensitivity.

## Methods

**Mice**. Animal protocols were approved by the Institutional Animal Care and Use Committee of Korea University. Mice were housed in a specific pathogen-free facility at Korea University. For muscle-specific TAZ-knockout mice, a floxed allele containing LoxP sites flanking exon 2 of *Taz* was generated. The first LoxP site was located upstream a neomycin selection cassette (Supplementary Fig. 1). The neomycin selection cassette was deleted by partial recombination through intercrossing with *EIIa-Cre* mice. Mice with *Taz* floxed alleles were crossed with muscle creatine kinase (MCK) *Cre* mice. In the presence of the *MCK-Cre* allele, the floxed exon 2 of *Taz* was excised by Cre-mediated recombination. To generate muscle-specific TAZ- and APC-double-knockout mice, *APC*^fl/fl^ mice were crossed with *MCK-Cre: Taz*^fl/fl^ mice. *APC*^fl/fl^ (C57BL/6-Apc^tm1Tyj^/J, #009045), *MCK-Cre* (FVB-Tg(Ckmm-cre)5Khn/J, #006405), and *EIIa-Cre* (FVB/N-Tg(EIIa-cre)C5379Lmgd/J, #003314) mice were purchased from Jackson Laboratory. For high-fat diet condition, 8-week-old mice were fed with diet from Research Diet (D12331) for 8 weeks.

**Simvastatin administration**. Simvastatin was prepared as a 4 mg per ml stock solution in 10% EtOH–0.1 N NaCl (pH 7.0). Six-week-old mice were administered simvastatin or vehicle through voluntary water consumption. Daily water consumption was measured using an indirect calorimetric chamber (Oxylet Systems),

and simvastatin was supplied in the drinking water at 40 mg per kg body weight daily. Analyses were performed after 3 weeks of administration.

**Insulin treatment**. Insulin was purchased from Sigma–Aldrich. For the in vitro assay, cells were seeded in culture plates and maintained in growth medium. After 24 h, cells were starved in serum-free DMEM for 3 h, treated with insulin, and harvested at the indicated time points. For the in vivo assay, mice were intraperitoneally injected with 1 U per kg (body weight) insulin. After 15 min, mice were sacrificed, and the gastrocnemius muscles were isolated, homogenised, and lysed for immunoblot analysis.

**Glucose tolerance test**. Mice were fasted for 16 h prior to the assay. D-glucose (2 g per kg body weight) was, then, administered intraperitoneally. Mice were bled by cutting their tail tips with a sterilised sharp knife. Blood glucose levels were measured using an ACCU-CHEK Active glucose meter (Roche) at the indicated time points.

**Insulin tolerance test**. Mice were fasted for 4 h, and insulin (1.25 U per kg body weight) was, then, administered intraperitoneally. As described for the glucose tolerance test, mouse tail tips were cut with a sterilised sharp knife to induce bleeding, and blood glucose levels were measured using an ACCU-CHEK Active glucose meter at the indicated time points.

**Immunofluorescence**. Mouse tissues were isolated, immediately soaked in 4% paraformaldehyde, and incubated at 4 °C, for 24–48 h. Fixed tissue was dehydrated in an ethanol series and embedded in paraffin wax (Leica) at 60 °C, overnight. Embedded tissue blocks were sectioned with a microtome (Leica, 5-μm thickness). Tissue sections were rehydrated in an ethanol series, and antigen retrieval was performed using a pressure cooker. Samples were blocked with 1.5% normal serum

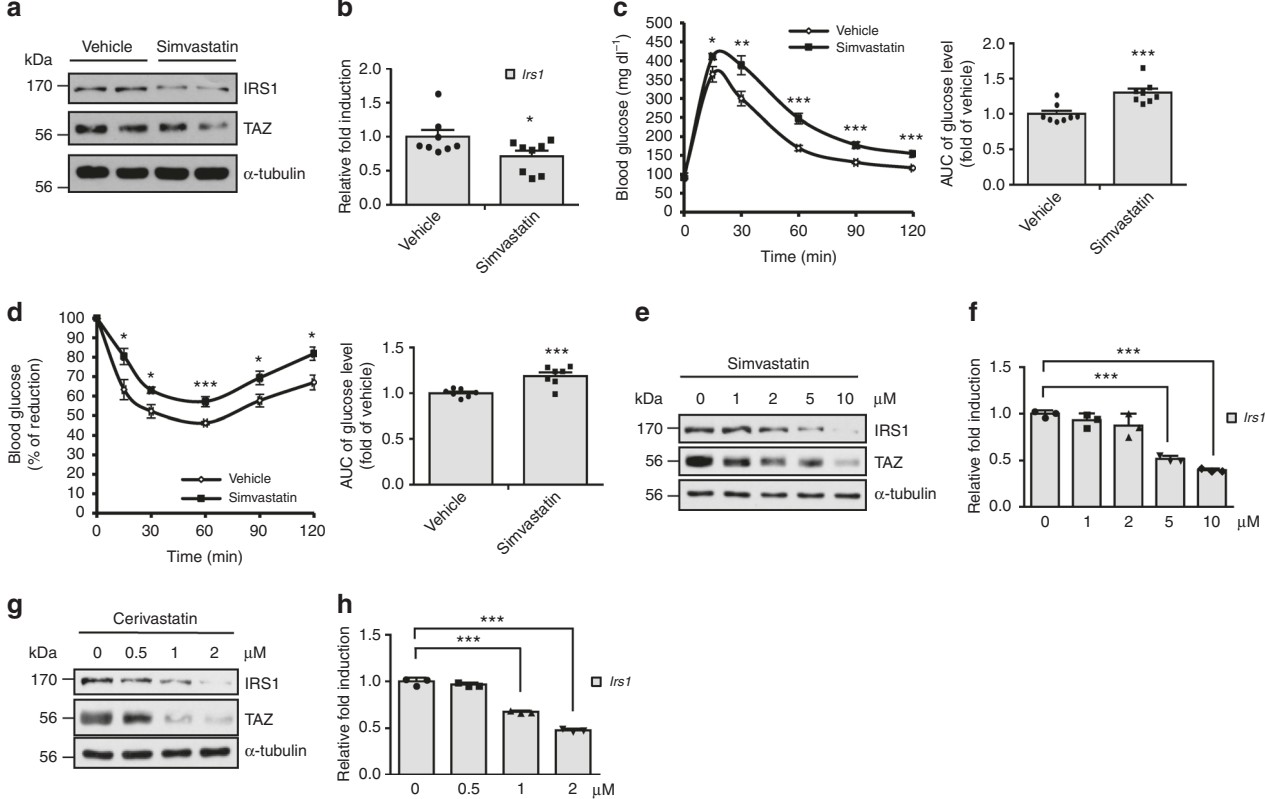

**Fig. 6** Statins decrease IRS1 and TAZ level, and insulin sensitivity. **a** Six-week-old mice were administered simvastatin or vehicle for 3 weeks. Mice were sacrificed, the gastrocnemius muscles were isolated, and IRS1 and TAZ level was analysed by immunoblotting. α-Tubulin was used as a loading control. **b** RNA was isolated from gastrocnemius muscles in panel **a**. *Irs1* transcription was analysed by qRT-PCR and normalised to *Gapdh* expression (*n* = 8). **c** Six-week-old mice were administered simvastatin or vehicle for 3 weeks. Mice were fasted for 16 h, and D-glucose was intraperitoneally injected at 2 g per kg body weight. Serum glucose levels were measured using a glucometer at the indicated time points. The area under the curve for each mouse was calculated and presented as a fold-change relative to that for the vehicle condition (*n* = 8). **d** Six-week-old mice were administered simvastatin or vehicle for 3 weeks. Mice were fasted for 4 h, then administered insulin at 1.25 U per kg body weight. Serum glucose levels were analysed at the indicated time points. The area under the curve for each mouse was calculated and presented a fold-change relative to that for the vehicle condition (*n* = 8). **e** C2C12 myotubes were treated with DMSO or simvastatin at the indicated concentration for 48 h, and cell lysates were analysed by immunoblotting. **f** Cells described in **e** were harvested, and *Irs1* expression was analysed by qRT-PCR and normalised to *Gapdh* expression (*n* = 3). **g** The same experiment in panel **e** was performed with cerivastatin instead of simvastatin. **h** The same experiment in panel **f** was assessed with cerivastatin instead of simvastatin. Data are presented as the relative fold induction (*n* = 3). Data are presented as mean ± SD for panels **f** and **h**, and as mean ± SEM for panels **b**–**d**. Statistical analysis was performed using a Student's *t*-test. *$p < 0.05$; **$p < 0.01$; ***$p < 0.005$

and incubated with specific primary antibodies diluted in PBS containing 0.05% Tween 20 (PBST), at 4 °C, overnight. The primary antibodies were removed by washing three times with PBST, and tissue sections were, then, incubated with fluorochrome-conjugated secondary antibodies diluted in PBST, at room temperature, for 2 h. After three washes with PBST, tissue sections were mounted using DAPI-containing mounting medium (Vector Laboratories). Fluorescent images were obtained by confocal microscopy (Carl Zeiss, LSM 510META).

**H&E staining**. Paraffin-embedded tissue sections were deparaffinised and rehydrated in an ethanol gradient. Sections were stained in Harris hematoxylin solution, differentiated in 1% acid alcohol, and incubated in 0.2% ammonia water for bluing. Counterstaining was performed in eosin Y solution, followed by dehydration. Finally, sections were mounted using a xylene-based mounting medium, and images were acquired using a light microscope (Nikon, ECLIPSE Ni-E) fitted with a digital camera (Canon, EOS 650D DS126371).

**Immunohistochemistry**. VECTORSTAIN Elite ABC HRP Kit (PK-6101, Vector Laboratories) was used for immunohistochemical analysis in accordance with the manufacturer's instructions. Tissue sections were deparaffinised and rehydrated in an ethanol gradient. Sections were exposed to 3% $H_2O_2$ to inhibit endogenous peroxidase activity. Thereafter, antigen retrieval of sections was performed in a pressure cooker in sodium citrate buffer. After washing with PBST (1× PBS with 0.05% Tween 20), sections were blocked with normal goat serum for 30 min. Primary antibody diluted in 2.5% normal goat serum-containing PBS was added to samples and incubated at 4 °C overnight. Sections were washed with PBST and incubated for 30 min in biotinylated secondary antibody diluted in 1.5% normal

goat serum containing PBS. After washing with PBST, sections were incubated in supplied reagent containing avidin DH and biotinylated peroxidase H for 30 min. Samples were then rinsed with PBS and stained with 3,3′-diaminobenzidine (DAB) peroxidase substrate kit (SK-4100, Vector Laboratories). To stain the nucleus, sections were counterstained with Mayer's hematoxylin solution. After clearing with an ethanol gradient and xylene, sections were observed using a light microscope (Nikon, ECLIPSE Ni-E).

**Oil-Red-O staining**. Mouse liver was isolated and embedded in OCT compound (VWR, 25608-930). After freezing at −20 °C, liver was sectioned using a cryotome (Leica, CM1950) at 5-μm thickness and mounted on a glass slide. Sectioned tissue was air-dried for 30 min and fixed in ice-cooled 3.7% formaldehyde for 10 min. Tissue was rinsed thrice with distilled water. After an additional air-drying step, tissue slides were immersed in absolute propylene glycol at 60 °C for 5 min. Tissue was stained with pre-warmed 0.5% Oil-Red-O solution (Sigma, O0625, dissolved in absolute propylene glycol) at 60 °C for 10 min. Thereafter, samples were differentiated in 85% propylene glycol for 5 min and rinsed twice with distilled water. Nuclei were counterstained with Mayer's hematoxylin (Sigma, MHS16) for 30 s. Tissue section was washed with running tap water for 3 min and mounted with aqueous mounting medium (DAKO, S3023).

**Serum preparation and analysis**. Blood was collected from fasted or fed mice via tail bleeding, incubated at room temperature for 30 min, and centrifuged at 1000×*g* at 4 °C. The supernatant was collected and stored at −80 °C for further analysis. Quantification of serum insulin (5 μl of serum per sample) was performed using a

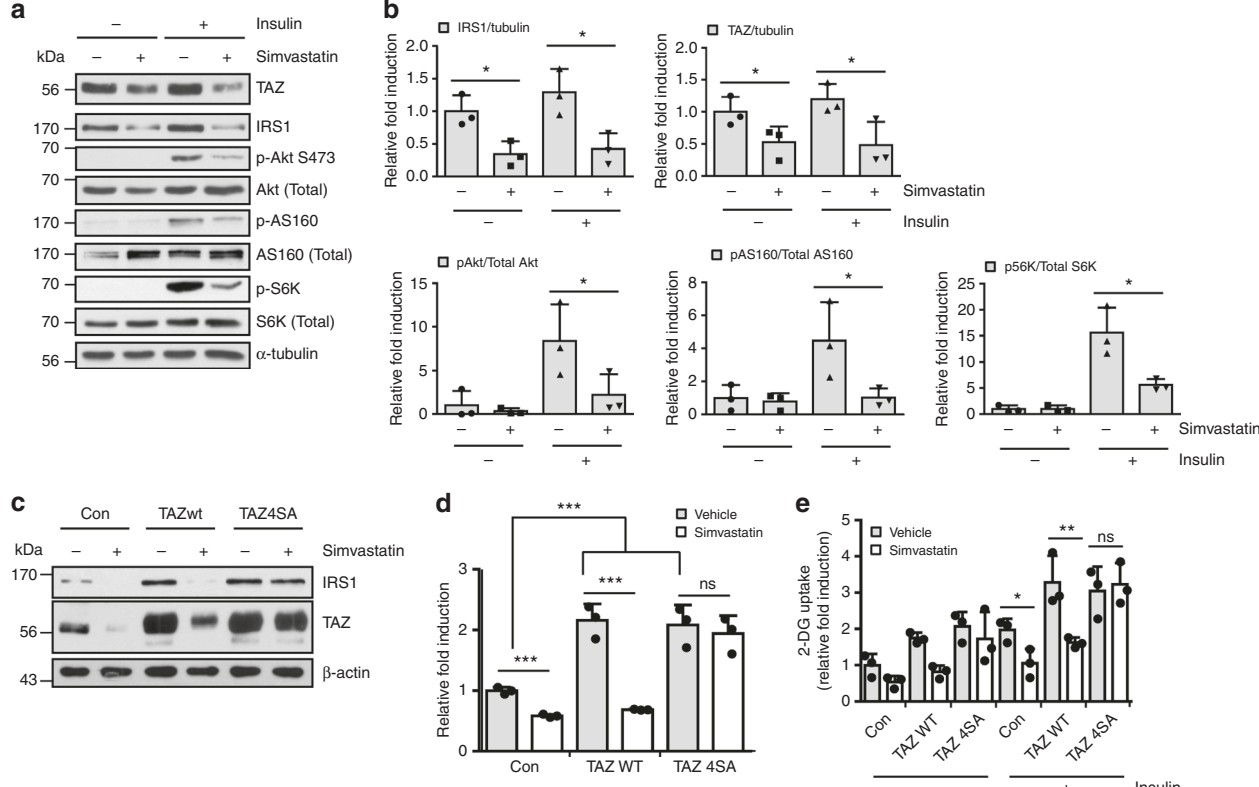

**Fig. 7** Statin treatment inhibits insulin signalling via TAZ degradation. **a** Insulin was intraperitoneally injected into vehicle- or simvastatin-administered mice. After 15 min, muscle lysates were analysed by immunoblotting. α-Tubulin was used as a loading control. Simvastatin or vehicle was administered to 6-week-old mice for 3 weeks. **b** Three independent experiments performed as described in **a** were assessed, and the ratios of IRS1 and TAZ to α-tubulin were analysed; in addition, the ratios of phosphorylated to total AKT, AS160, and ribosomal protein S6 kinase were determined ($n = 3$). **c** C2C12 cells were transduced with control, wild-type TAZ, or proteolytic degradation-resistant TAZ (TAZ4SA; TAZ S58, 62, 306, 309A) retroviral vectors to establish stable cell lines. Then, 10 μM simvastatin or DMSO was added to cells for 48 h, and IRS1 and TAZ level in harvested cells was analysed by immunoblotting. **d** *Irs1* expression in the cells described in **c** was analysed by qRT-PCR and normalised to *Gapdh* expression ($n = 3$). **e** Glucose uptake was analysed in control, wild-type TAZ, and TAZ4SA-overexpressing C2C12 myotubes ($n = 3$). Data are presented as mean ± SD. Statistical analysis was performed using a Student's *t*-test. ns not significant; *$p < 0.05$; **$p < 0.01$; ***$p < 0.005$

Mouse Insulin ELISA Kit (Insulin: ALPCO, 80-INSMSU-E10) according to the manufacturer's instructions.

**Antibodies**. Anti-IRS1 (#2382), anti-p-IRS1 S307 (#2381), anti-IRS2 (#4502), anti-p-Akt T308 (#9275), anti-p-P70S6K (#9234), anti-P70S6K (#2708), anti-p-AS160 (#8730), anti-AS160 (#2670), non-phospho (active) anti-β-catenin (#8814), anti-Vinculin (#13901), and anti-TAZ/YAP (#8418) antibodies were purchased from Cell Signaling Technology. Anti-Akt (sc-8312), anti-p-Akt S473 (sc-101629), anti-IRα (sc-710), anti-c-Jun (sc-74543 for immunoprecipitation and ChIP and sc-1694 for immunoblotting), anti-YY1 (sc-281), anti-C/EBPα (sc-61), and anti-α-tubulin (sc-5286) antibodies were purchased from Santa Cruz Biotechnology. Anti-Glut4 antibody (#2203-1) was purchased from Epitomics. Anti-IR β antibody (07-724) was obtained from Millipore. Anti-FLAG (F1804) and TAZ (HPA007415) antibodies were purchased from Sigma–Aldrich. Anti-Tead4 antibody (ab97460 for immunoblotting and ab58310, for immunoprecipitation) and anti-IRS1 antibody for immunohistochemistry (ab52167) were purchased from Abcam. Anti-TAZ antibody for ChIP (NB110-58359) was purchased from Novus Biologicals. For the immunoblot assay, antibodies were diluted to a working concentration of 1:1000 for anti-IRS1, anti-p-IRS1 S307, anti-IRS2, anti-IRα, anti-IRβ, anti-c-Jun, anti-YY1, anti-C/EBPα, anti-GLUT4, and anti-Tead4 antibodies, 1:2000 for anti-p-Akt T308, anti-p-Akt S473, anti-Akt, anti-p-P70S6K, anti-P70S6K, anti-p-AS160, anti-AS160, anti-YAP/TAZ, anti-α-tubulin, anti-FLAG, and anti-TAZ antibodies, and 1:5000 for anti-β-actin antibody. For immunohistochemistry, anti-TAZ and anti-IRS1 antibodies were diluted to a working concentration of 1:100. For co-immunoprecipitation, anti-TAZ (NB110-58359) and anti-FLAG antibodies were diluted to 1:100 and 4 μg per mg of protein for anti-c-Jun (sc-74543) and anti-Tead4 (ab58310) antibodies. For ChIP, antibodies were diluted to 0.5 μg per μg of DNA for anti-TAZ, anti-c-Jun, and anti-FLAG antibodies. Uncropped immunoblots of all figures are shown in Supplementary Fig. 11.

**Cell culture**. C2C12 cells (ATCC) were maintained in DMEM containing 20% fetal bovine serum. MEFs were prepared from mouse embryos and maintained in

DMEM containing 10% fetal bovine serum. Both growth media were supplemented with penicillin and streptomycin. For myogenic differentiation, C2C12 cells were seeded in culture plates and maintained for 24 h. When cells reached confluency, the medium was changed to DMEM containing 2% horse serum to promote differentiation. All cells were incubated at 37 °C, in a 5% $CO_2$ atmosphere.

**Glucose uptake assay**. C2C12 myoblasts were seeded in a 96-well culture plate and differentiated into myotubes for 3 days. After 3 h of serum starvation in serum-free DMEM, MEFs and C2C12 myotubes were starved for glucose by incubation with Krebs-Ringer-phosphate-HEPES buffer (containing 2% BSA), for 40 min. Then, cells were stimulated with 100 nM insulin for 20 min and incubated with 10 μl of 10 mM 2-deoxyglucose, at 37 °C, for 20 min. To analyse glucose uptake, a Glucose Uptake Colorimetric Kit (Biovision, K676-100) was used, according to the manufacturer's instructions. At the end of the reaction, the absorbance of the samples and standards was measured at 412 nm, using a microplate reader (Bio-Rad).

**Cloning of IRS1 enhancer luciferase reporter gene constructs (TBE1-luc)**. Using PCR, mouse genomic DNA was amplified with primers spanning the IRS1 enhancer region (TBE1). Primers contained restriction enzyme sites at their 5′ ends (*Kpn*I in the forward primer and *Hind*III in the reverse). PCR products and pGL3 basic vector were digested with *Kpn*I and *Hind*III for ligation. After transformation, plasmids were prepared and used for the luciferase reporter gene assay. Primers used in the cloning are listed in Supplementary Table 1.

**Site-directed mutagenesis**. Primers containing a mutated c-Jun-binding sequence were designed to convert the original genomic sequence CGAAGTGTCCTCCTC**ATGACTT**TTCGTAAATAAGAAG to CGAAGTGTCCTCCTCA**TTTTTT**TTC GTAAATAAG (nucleotides in bold indicate the mutated c-Jun DNA-binding sequence). The TBE1-luc construct was used as the PCR mutagenesis template. Following PCR, 10 U of *Dpn*I were added to digest the parental template plasmid

strand. The products were transformed into DH5α competent cells and incubated on an ampicillin-containing agar plate for mutant plasmid selection. Finally, pre-prepared plasmids were sequenced to confirm the presence of the mutation in the c-Jun DNA-binding sequence.

**Retroviral transduction to generate stable cell lines**. Phoenix cells were transfected with retroviral vectors and viral packaging vectors, using the calcium phosphate precipitation method. Virus-containing medium was harvested 24 h after transfection, filtered through a 0.45-μm filter, and added to target cells together with 4 μg per μl polybrene. To select for transfected cells, cells were cultured in growth medium containing 4 μg per ml puromycin (for pBabe-puro and pSRP) or 100 μg per ml zeocin (for pBabe-bleo) for 1–2 weeks. For TAZ knockdown, pSRP-mTAZ plasmid (Addgene plasmid #31795) was used. In case of wild type TAZ-overexpression, pBabe puro-mTAZ plasmid (Addgene plasmid #31791) was used. For IRS1-rescued cell line, pBabe bleo-hIRS1 plasmid (Addgene plasmid #11448) was used.

**RNA interference for transient knockdown**. C2C12 myoblasts were seeded in 6-well culture plates, at a density of $1 \times 10^5$ cells per well. After 24 h, cells were transfected with siRNA for c-Jun, YY1, C/EBPα, or Tead4, or with the corresponding scrambled siRNA, using Lipofectamine 2000 transfection reagent (Invitrogen, cat #11668019). Cells were incubated for 48 h and harvested for immunoblotting or quantitative real-time (qRT) PCR. Target sequences for RNA interference are presented in Supplementary Table 1.

**Co-immunoprecipitation**. Cells were harvested in ice-cold PBS and lysed in RIPA buffer [150 mM NaCl, 50 mM Tris–HCl (pH 7.4), 1 mM EDTA, 1% NP-40, 0.5% sodium deoxycholate, 0.1% SDS, 1 mM $Na_3VO_4$, 1 mM NaF] containing protease inhibitors. Total proteins (0.5–1.0 mg) were pre-cleared with BSA-coated Sepharose beads (Sigma–Aldrich). Cleared lysates were incubated with specific antibodies at 4 °C, overnight, with gentle rotation. Antigen–antibody complexes were precipitated with protein G-Sepharose beads (Sigma–Aldrich) at 4 °C, for 4–6 h. Proteins were eluted with 2× SDS sample buffer by boiling at 98 °C, for 5 min.

**Chromatin immunoprecipitation (ChIP)-PCR and ChIP sequencing**. In total, $4 \times 10^6$ cells were seeded in 150-cm² plates and maintained in growth medium for 24 h. For ChIP sequencing, FLAG-TAZ-expressing C2C12 cells were used to capture TAZ-bound chromatin fragments. Formaldehyde was, then, added to cells at a final concentration of 0.75% for crosslinking, followed by the addition of 125 mM glycine to quench the crosslinking reaction. After three washes in ice-cold PBS, cells were collected in ice-cold PBS and centrifuged. Pellets were lysed with FA lysis buffer [50 mM HEPES-KOH (pH 7.5), 140 mM NaCl, 1 mM EDTA (pH 8.0), 1% Triton X-100, 0.1% sodium deoxycholate, 0.1% SDS, protease inhibitor cocktail] and sonicated using a Bioruptor sonicator (Diagenode). Chromatin was sheared into 500–1000- and 200–300-bp fragments for ChIP-PCR and ChIP sequencing, respectively. The DNA concentrations of the fragmented samples were measured, and 20 μg of sheared chromatin were used for immunoprecipitation. To capture TAZ-bound DNA fragments, anti-FLAG M2 agarose beads (Sigma, A2220) coated with BSA and salmon sperm DNA were added to samples and incubated at 4 °C, for 4 h. For c-Jun-bound chromatin regions, c-Jun antibodies were added to samples and incubated at 4 °C, for 16 h. Protein G beads coated with BSA and salmon sperm DNA were used to capture c-Jun-chromatin complexes. After three washes with wash buffer [0.1% SDS, 1% Triton X-100, 2 mM EDTA (pH 8.0), 150 mM NaCl, 20 mM Tris–HCl (pH 8.0)], samples were washed with the final wash buffer [0.1% SDS, 1% Triton X-100, 2 mM EDTA (pH 8.0), 500 mM NaCl, 20 mM Tris–HCl (pH 8.0)] and eluted with 1% SDS, 100 mM NaHCO₃, at 30 °C, for 15 min, with gentle agitation. Eluted samples were reverse crosslinked by incubation with RNase A, at 65 °C, for 5 h, with gentle rotation. DNA samples were purified using a Gel Extraction/PCR Purification Kit (Thermo). Purified samples were analysed by qRT-PCR (LC480, Roche) for ChIP-PCR or sequenced with NGS technology for bioinformatics analysis. For ChIP-PCR, immunoprecipitated DNA was quantified as the percent of the input fraction, using the following formula: amplification efficiency^(Ct Input − Ct ChIP). The resulting values were normalised to those of the IgG or empty vector-infected control. Primers for qRT-PCR are presented in Supplementary Table 1.

**Analysis of ChIP sequencing data**. Purified ChIPed samples were sequenced using NGS technology. Cutadapt (version 1.8) was used to trim adapter sequences, and BWA mapper was used to map the remaining reads to the mouse genome (mm10). Peak regions were predicted using MACS (version 2), and peak annotations were performed using HOMER. The IGV genome browser was used to visualise sequencing reads. Chromatin modifications (H3K27ac, H3K4 me1, and H3K4 me3) were visualised in raw ChIP sequencing data reads downloaded from the NCBI Sequence Read Archive (SRP009088). For motif analysis, approximately 50 bases flanking each peak were analysed using the PROMO virtual laboratory program (using TRANSFAC 8.3 version) and JASPAR.

**Gene expression analysis**. Total RNAs were prepared using TRIzol reagent (Invitrogen). Reverse transcription was performed using M-MLV reverse transcriptase (Thermo). A LightCycler480 (Roche) system was used to analyse mRNA expression levels via qRT-PCR. For relative quantification, the transcript levels of *Gapdh* were assessed in every condition as a reference value, and the ratios of reference to target genes were calculated with the following formula: amplification efficiency^(Ct reference − Ct target). Calculated values were normalised to those of the control condition. Forward and reverse primer sequences are presented in Supplementary Table 1.

**Luciferase reporter gene assay**. HEK293T cells were transfected with TBE1-luc plasmids with or without TAZ, c-Jun, and Tead4 expression vectors. A *Renilla* luciferase plasmid was used as transfection control. Xtremegene 9 transfection reagent (Roche) was used for the transfection procedure, according to the manufacturer's instruction. Cells were lysed after 24 h, and firefly and *Renilla* luminescence was measured using a Dual Luciferase Kit (Luciferase assay system, *Renilla* luciferase assay system, Promega). Luminescence was detected using a luminometer (GLOMAX, Promega), and firefly luciferase activity was normalised to that of *Renilla* luciferase.

**Cell fractionation for isolation of plasma membrane proteins**. C2C12 myotubes in two 150-cm² dishes were serum-starved, for 3 h, in serum-free DMEM and, then, treated with 100 nM insulin, for 30 min. Cells were washed, harvested in ice-cold PBS, and centrifuged. After removing the supernatant, the plasma membrane and cytosolic fractions were isolated using a Membrane Protein Extraction Kit (Biovision). All fractionation procedures were performed according to the manufacturer's instructions.

**Statistical analysis**. All in vivo data are presented as the mean and standard error of the indicated number of experimental samples. In case of in vitro analyses, data are shown as the mean and standard deviation of at least three independent experimental sets. Statistical significance was calculated using Student's *t*-test, and the significance levels are indicated with asterisks as follows: *$p < 0.05$; **$p < 0.01$; ***$p < 0.005$; ns, not significant.

**Reporting Summary**. Further information on experimental design is available in the Nature Research Reporting Summary linked to this Article.

## Data availability

ChIP sequencing data have been deposited in the ArrayExpress database under accession code E-MTAB-6764. All other data of this study are available from the corresponding authors upon reasonable request.

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

## Acknowledgements

This work was supported by the Basic Science Research Program of NRF funded by the Korea Government (2015R1A5A1009024, 2018R1A5A2025286, and 2018R1D1A1A09037028). This work was also supported by a Korea University grant.

## Author contributions

J.-H. Hwang performed the experimental work, analysed the data, and wrote the manuscript. A.R.K., K.M.K., J.I.P., H.T.O., S.A.M., M.R.B., H.J and H.K.K. prepared the experimental reagents, set up the experimental system, and contributed to discussions of the data. M.B.Y. read the manuscript and provided feedback. E.S.H. and J.-H. Hong led the project, interpreted the data, and wrote the manuscript.

## Additional information

**Competing interests:** The authors declare no competing interests.

