## [Peer Review File · Nature Communications]

Reviewers' comments:

Reviewer #2 (Remarks to the Author):

The manuscript by Hwang et al addresses the role of Taz protein in regulating glucose uptake and insulin sensitivity in skeletal muscle. To this end the authors generated a muscle specific Taz-null mice lacking Taz exclusively in mature muscle fibres and they investigated the insulin downstream signalling pathways. They show that upon stimulation with insulin there is an impaired Akt activity, lower Glut4 levels in the plasma membrane and decreased Irs1 expression in Taz-KO muscles, Taz-null MEFs and Taz-null C2C12 cells. Then they performed standard biochemistry and molecular biology analysis on cell lines, showing that Taz interacts with Jun and binds to Irs1 promoter, regulating its transcription. Finally, they used C2C12 cells and APC/Taz double KO mice to demonstrate that Taz mediates Wnt signalling-induced activation of Irs1. Through the above experiments they conclude that Taz1 mediates glucose uptake in skeletal muscle through upregulation of Irs1.

The study addresses an interesting issue and it is of potential interest in the field. However, the results are largely correlative and insufficient to support the main conclusions made by the authors. The observation that Yap/Taz regulates insulin sensitivity through regulation of Irs1 was previously observed in endometrial cancer (Wang et al, Am J Cancer Res. 2016), which diminish some of the novelty of the current work. More substantial work is needed before this study meets the standard of Nature Communications. In particular, a more detailed phenotypic characterization of the Taz-null mice (seem below) and formal proof that Taz regulates glucose uptake through up-regulation of Irs1 should be provided.

Major points

- 1) A more detailed characterization of the Taz-KO mice phenotype should be given. Do muscle specific Taz-KO mice develop any metabolic disorder? From data in supplementary figure 4 it seems that Taz-depletion in muscle fibres does not alter muscle morphology, but those results are based only on immunofluorescence analysis of ~~glycogen~~ ^{glycogen} and no quantification of the cross-sectional area is provided. In addition, the authors should analyse if there is a switch of fibre composition (slow vs fast) in KO mice due to impaired glucose uptake. Further analysis of the muscle-specific Taz-null mice at different ages and in response to changes in diet would help to clarify the consequences of impaired glucose uptake in Taz-null skeletal muscle.
- 2) The authors state that Taz regulates glucose uptake through chromatin binding and transcriptional regulation of Irs1. However, they fall short in presenting any mechanistic evidence showing that down-regulation of Irs1 mediates Taz-null mice deficiency. To this end the authors should perform rescue experiments to show that Irs1 over-expression rescues the Taz-KO phenotype. Moreover, recent work showed that Yap/Taz regulates insulin sensitivity in endometrial cells through post-transcriptional regulation of Irs1 (Wang et al, Am J Cancer Res. 2016). Is this mechanism also conserved in skeletal muscle?
- 3) The authors performed ChIP-seq analysis using flag-tagged Taz on C2C12 cells. Are those proliferating or differentiated cells? Are they stimulated with insulin? The ChIP-seq data should be further analyzed, including a control input and the raw data should be made available through the relevant databases. Does Taz mainly binds to promoters, gene bodies or intergenic regions? How

many Taz targets were identified and why did the author focused on Irs1 only? Do Taz chromatin localization change upon insulin treatment? Previous work by the authors showed that Taz interacts with and activates the muscle regulatory factor MyoD in differentiated muscle cells. Did the CHIP-seq analysis confirmed previous data by showing Taz enrichment at MyoD targets? Finally, CHIP data should be validated also in the mouse model, before and after insulin infusion.

4) A histological analysis of Taz and Irs1 localization in skeletal muscle sections should be performed in basal conditions and upon insulin infusion.

5) In figure 4b-c the authors show co-immunoprecipitation analysis of Taz-Jun interaction in 293T and C2C12 cells. However, in order for this interaction to be relevant in vivo the authors should perform the Co-IP using skeletal muscle extracts. Similar experiments should be also performed to analyse the presence of Tead4 in the complex, as siRNA-mediated depletion of this factor showed it cooperates with Taz and Jun to regulate Irs1 transcription.

Minor points

1) The word "expression" refers to genes, not to proteins. The authors should use the word "levels" when referring to proteins to avoid confusion and to help to identify if they refer to transcriptional or post-transcriptional mechanisms.

2) In figure 3a, the "highest reads score" is not a measure of strength of binding. Proper analysis should be performed for the CHIP-seq experiments.

3) CHIP analysis in figures 3b and 4a would be better represented as percent of input, instead of relative to IgG condition, to indicate the amount of binding at each condition. A negative control should be also included in those experiments.

4) The cross-sectional area in supplementary fig. 4 should be quantified.

5) The age of the animals and the muscle analysed should be specified in all experiments

Reviewer #3 (Remarks to the Author):

Hwang et al examine here the role of TAZ/YAP in insulin signalling. Specifically, they found that TAZ is required for basal IRS1 expression and that Wnt-induced insulin sensitivity is mediated through TAZ-induced increase in IRS1 mRNA in skeletal muscle. TAZ was found to act in association with c-Jun and TEAD in order to regulate IRS1 mRNA levels. Second, statin-induced insulin resistance was found to be associated with decreased TAZ and IRS1 expression, while overexpression of TAZ rescued statin-induced insulin resistance.

Strengths:

1. This study establish a new connection between two previously known facts: that Wnt signalling induces IRS1 expression and that Wnt along with nutrient/metabolic sensing pathways regulate TAZ. Thereby, it establishes TAZ as a novel regulator of insulin signalling. The novelty of this work lies in the identification of TAZ-c-Jun and TAZ-TEAD mediated IRS1 expression as well as in the determination of TAZ as a mediator of statin-induced insulin resistance.

2. The experiments performed are clear and methodical in approach, and the findings are convincing. The use of a number of models strengthens the authors' findings, and supportive measures are provided to strengthen each conclusion.

Weaknesses:

1. The study does not provide deep mechanistic insight, it is rather a signalling study which, although very well performed, is not of high cell biology novelty in approach.

2. The physiological context of Wnt regulating IRS1 is not provided.

3. Statins are known to cause insulin resistance through other pathways (including reducing prenylation of small GTPases). These mechanisms must be discussed vis a vis the current results. Do the authors ascribe the statin-induced insulin resistance solely to changes in IRS1?

Other Issues:

1. On line 92, when referring to figure 2d, the authors state that glucose clearance is attenuated by TAZ KO. However, it appears that the peak glucose concentration in blood was higher in the KO, while the rate of clearance remains the same. Therefore, the rate of clearance does not appear in fig 2d to be altered but rather the initial response and/or the glucose load was higher in the KO. The rate of glucose clearance could be examined using Ki.

2. To ensure that the effects of TAZ are limited to IRS1 expression alone, the authors should examine whether AKT phosphorylation can be rescued by IRS1 overexpression in TAZ KO MEFs or TAZ siRNA treated C2C12.

3. Fig 5c: the representative image for IRS1 in the knockdown does not match the quantification in Fig 5d. IRS1 is increased in 5c knockdown treated with Wnt3a compared to knockdown untreated but the quantification shows no increase with low variation between repeat experiments.

4. For APCmKO mice the authors should present control measures of Wnt signalling to show that the KO displays the desired effects of increased Wnt signalling.

5. The introduction is scant and lacks biological context.

6. Fig. 2b shows results of Glut4 in C2C12 myotubes. This is surprising as this cell line is notorious for little or no expression of the transporter even in the myotube stage. Perhaps a validation of the antibody would be useful to strengthen the finding? Second, myoblasts certainly do not express Glut4, and yet glucose uptake was measured in myoblasts (2a). This is confusing and should be done in myotubes.

Responses to Reviewer 2's Comments:

1) A more detailed characterization of the Taz-KO mice phenotype should be given. Do muscle specific Taz-KO mice develop any metabolic disorder? From data in supplementary figure 4 it seems that Taz-depletion in muscle fibres does not alter muscle morphology, but those results are based only on immunofluorescence analysis of β -dystroglycan and no quantification of the cross-sectional area is provided. In addition, the authors should analyse if there is a switch of fibre composition (slow vs fast) in KO mice due to impaired glucose uptake. Further analysis of the muscle-specific Taz-null mice at different ages and in response to changes in diet would help to clarify the consequences of impaired glucose uptake in Taz-null skeletal muscle.

Response: Upon administering standard laboratory chow, we did not observe any metabolic disorder except for a difference in insulin sensitivity. No phenotypic differences were observed in liver and fat tissue. In a high-fat diet, however, we observed several phenotypic differences. TAZ mKO mice showed decreased insulin sensitivity and ability of glucose disposal (Supplementary Figure 5a and 5b). Increased body weight of mKO mice was observed with increased adipocyte size (Supplementary Figure 5c and 5d). Interestingly, mKO mice developed a severe fatty liver phenotype evident from increased fat droplets in liver sections (Supplementary Figure 5e). Thus, these results suggest that TAZ is an important regulator of insulin signalling in muscles, thereby influencing whole-body metabolism.

Supplementary Figure 5

Muscle fibre type composition was analysed on the basis of the expression of marker genes of each fibre type (Type I for slow, Type II for fast), using quantitative real-time PCR. There were no significant differences in muscle fibre type composition between WT and mKO muscle (Supplementary Figure 4d). The cross-sectional area of muscle fibres was measured and a significant difference was not observed between WT and mKO muscle tissues (Supplementary Figure 4b). Muscle weight was also similar in WT and mKO mice (Supplementary Figure 4c)

Supplementary Figure 4

2) The authors state that Taz regulates glucose uptake through chromatin binding and

transcriptional regulation of *Irs1*. However, they fall short in presenting any mechanistic evidence showing that down-regulation of *Irs1* mediates *Taz*-null mice deficiency. To this end the authors should perform rescue experiments to show that *Irs1* over-expression rescues the *Taz*-KO phenotype. Moreover, recent work showed that *Yap/Taz* regulates insulin sensitivity in endometrial cells through post-transcriptional regulation of *Irs1* (Wang et al, Am J Cancer Res. 2016). Is this mechanism also conserved in skeletal muscle?

Response: As suggested, the rescue experiments were assessed. Rescued *IRS1* in *TAZ* KO MEFs restored *AKT* activation (Supplementary Figure 3c) and glucose uptake (Supplementary Figure 3d). In our study, *TAZ* depletion significantly decreased *IRS1* levels in muscle tissue, MEF cells, and C2C12 cells. In addition, *TAZ* overexpression rescued *IRS1* levels with a significant induction of *Irs1* expression. Thus, *TAZ* plays an important role in *Irs1* expression, at least in the muscle tissue, C2C12 myoblasts, and MEFs.

Supplementary Figure 3

As the reviewer suggested, Wang et al. reported that IGF1 increases *IRS1* levels, accompanied with phosphorylation of *IRS1* at serine 307 and depletion of *TAZ* and *YAP* decreases *IRS1* levels in endometrial cancer cell lines. We performed a similar experiment in WT and *TAZ*-depleted C2C12 myoblasts (bottom Supplementary Figure 10). We observed that *IRS1* phosphorylation at Ser 307 was increased in control cells (con, scrambled siRNA treated) and *TAZ* knockdown (Ti, *TAZ* specific siRNA treated) cells after 5 min of insulin treatment. However, we did not observe a significant increase in *IRS1* levels of both cells. Thus, post-transcriptional regulation of *IRS1* does not play a major role in insulin-mediated

IRS1 induction in C2C12 myoblasts. Notably, the basal level of IRS1 was significantly decreased upon TAZ knockdown cells in the absence of insulin. Thus, based on our other results, *Irs1* expression contributes to basal production of IRS1 in myocytes.

Supplementary Figure 10. Control and TAZ knockdown C2C12 myoblasts were serum-starved for 3 h with serum-free DMEM and 1 nM insulin was administered to the cells. Thereafter, cells were harvested at the indicated time points and analysed via an immunoblot assay. β -actin was used as a loading control.

3) The authors performed ChIP-seq analysis using flag-tagged Taz on C2C12 cells. Are those proliferating or differentiated cells? Are they stimulated with insulin? The ChIP-seq data should be further analyzed, including a control input and the raw data should be made available through the relevant databases. Does Taz mainly binds to promoters, gene bodies or intergenic regions? How many Taz targets were identified and why did the author focused on *Irs1* only? Do Taz chromatin localization change upon insulin treatment? Previous work by the authors showed that Taz interacts with and activates the muscle regulatory factor MyoD in differentiated muscle cells. Did the ChIP-seq analysis confirmed previous data by showing Taz enrichment at MyoD targets? Finally, ChIP data should be validated also in the mouse model, before and after insulin infusion.

Response: We used proliferating C2C12 myoblasts for ChIP-seq analysis; these cells were not treated with insulin. The raw data are accessible through the ArrayExpress database (<https://www.ebi.ac.uk/arrayexpress/>). For reviewer access, please use the following login details:

Username: Reviewer_E-MTAB-6764

Password: Ea11vMzY

Upon assessment of peak annotation, we observed that numerous TAZ-binding sites were located in introns or intergenic regions, not in the promoter or exon regions. We could identify 23,804 targets via ChIP-seq analysis (bottom Supplementary Figure 11).

Supplementary Figure 11. Annotation of whole ChIP-seq peaks were assessed, and the distribution of annotation is shown as a pie graph (TSS: transcription start site; TTS: transcription termination site).

In this study, we initially investigated whether known insulin signalling components are regulated by TAZ and observed that *Irs1* expression was regulated by TAZ (Figure 1e). Thus, we attempted to identify cis-regulatory elements for the regulation of *Irs1* expression via ChIP-seq analysis. Therefore, the primary purpose of ChIP-seq analysis was the identification of *Irs1* gene regulatory elements at which TAZ binds. Along with *Irs1*, we verified TAZ binding to the regulatory region of known TAZ target gene *Cyr61* and *Myogenin* (bottom Supplementary Figure 12).

Supplementary Figure 12. ChIP sequencing peaks flanking the well-known TAZ target gene (*Cyr61*) and MyoD target gene (*Myog*) were visualized using the IGV genome browser with reads for histone modification status (H3K27ac, H3K4me3, and H3Kme1). Scale bar = 500 base pairs.

In the mouse model, we observed that TAZ localized at the TBE1 of *Irs1* in the absence of the insulin signal, which was not observed in mKO mice. After 15 min of insulin treatment, a slight increase of TAZ binding at the TBE1 was observed (Supplementary Figure 7). These results suggest that TAZ is important for basal *Irs1* expression.

Supplementary Figure 7.

4) A histological analysis of Taz and Irs1 localization in skeletal muscle sections should be performed in basal conditions and upon insulin infusion.

Response: We performed immunohistochemical analysis for TAZ and Irs1 in the skeletal muscle under basal and insulin-stimulated condition. TAZ primarily localized to the nucleus in both conditions and IRS1 was distributed mostly in the cytoplasm. We did not observe a significant difference in TAZ and IRS1 localization between basal and insulin-treated conditions (bottom Supplementary Figure 13).

Supplementary Figure 13. Vehicle- or insulin- (1 U/kg body weight) treated WT mice were euthanised and the gastrocnemius muscle was dissected out, fixed, dehydrated, and embedded

in paraffin. Sectioned tissues were analysed via immunohistochemistry with anti-TAZ and anti-IRS1 antibodies to determine the level and localization of each protein. Scale bar = 50 μ m

5) In figure 4b-c the authors show co-immunoprecipitation analysis of Taz-Jun interaction in 293T and C2C12 cells. However, in order for this interaction to be relevant *in vivo* the authors should perform the Co-IP using skeletal muscle extracts. Similar experiments should be also performed to analyse the presence of Tead4 in the complex, as siRNA-mediated depletion of this factor showed it cooperates with Taz and Jun to regulate Irs1 transcription.

Response: We obtained co-immunoprecipitation data on mouse skeletal muscle. TAZ and c-Jun or Tead4 interactions were confirmed *in vivo* (Supplementary Figure 8a and 8b).

Supplementary Figure 8.

Minor points

1) The word “expression” refers to genes, not to proteins. The authors should use the word “levels” when referring to proteins to avoid confusion and to help to identify if they refer to transcriptional or post-transcriptional mechanisms.

Response: Thank you for your comments. We revised ‘expression’ to ‘levels’.

2) In figure 3a, the “highest reads score” is not a measure of strength of binding. Proper analysis should be performed for the ChIP-seq experiments.

Response: We predicted ChIP-seq peaks using Model-based Analysis of ChIP-Seq (MACS) algorithm with a default p-value cut off (10^{-5}). We considered the peak scores derived from

the MACS algorithm and histone modification status data available in the GEO database to select major peaks to be analysed further. With these two data sets, TBE1 was selected as the primary target peak for further identification.

3) ChIP analysis in figures 3b and 4a would be better represented as percent of input, instead of relative to IgG condition, to indicate the amount of binding at each condition. A negative control should be also included in those experiments.

Response: Thank you for your comments. We revised the ChIP-qPCR data to represent a percentage of the input and a non-peak negative control was included in Figure 4a.

Figure 4.

4) The cross-sectional area in supplementary fig. 4 should be quantified.

Response: As suggested, the cross-sectional area was quantified along with staining data (Supplementary Figure 4b).

Supplementary Figure 4.

5) The age of the animals and the muscle analysed should be specified in all experiments.

Response: As suggested, the age of the all animals used was specified in the figure legends.

Responses to Reviewer 3's Comments:

Hwang et al examine here the role of TAZ/YAP in insulin signalling. Specifically, they found that TAZ is required for basal IRS1 expression and that Wnt-induced insulin sensitivity is mediated through TAZ-induced increase in IRS1 mRNA in skeletal muscle. TAZ was found to act in association with c-Jun and TEAD in order to regulate IRS1 mRNA levels. Second, statin-induced insulin resistance was found to be associated with decreased TAZ and IRS1 expression, while overexpression of TAZ rescued statin-induced insulin resistance.

Strengths:

1. This study establish a new connection between two previously known facts: that Wnt signalling induces IRS1 expression and that Wnt along with nutrient/metabolic sensing

pathways regulate TAZ. Thereby, it establishes TAZ as a novel regulator of insulin signalling. The novelty of this work lies in the identification of TAZ-c-Jun and TAZ-TEAD mediated IRS1 expression as well as in the determination of TAZ as a mediator of statin-induced insulin resistance.

2. The experiments performed are clear and methodical in approach, and the findings are convincing. The use of a number of models strengthens the authors' findings, and supportive measures are provided to strengthen each conclusion.

Weaknesses:

1. The study does not provide deep mechanistic insight, it is rather a signalling study which, although very well performed, is not of high cell biology novelty in approach.

Response: As the reviewer suggested, in our study, we described a novel insulin signalling regulatory pathway involving TAZ and proposed that TAZ links Hippo/Wnt signalling and insulin sensitivity. We observed that TAZ upregulates IRS1 and stimulates Akt and Glut4-mediated glucose uptake in myocytes. In muscle-specific TAZ-knockout (mKO) mice, IRS1 expression and insulin sensitivity were significantly decreased. In addition, the present study reported that statin-derived alterations in insulin sensitivity are mediated by TAZ. Thus, the present results suggest that TAZ downregulation may be a putative cause of statin-driven diabetogenic activity. In addition, we included the phenotype of mKO mice upon administration of a high-fat diet, in the revised manuscript. Interestingly, we observed that mKO mice have increased body weight, adipocyte size, and fatty liver phenotype (Supplementary Figure 5), suggesting that TAZ plays an important role in cellular metabolism via regulation of insulin sensitivity. We hope that these additional data improve the quality of our manuscript.

Supplementary Figure 5.

2. The physiological context of Wnt regulating IRS1 is not provided.

Response: Thank you for your comment. We have included the following contents in the Results section; “Insulin signalling cross-talks with the Wnt signal and altered Wnt signalling components impair glucose metabolism and diabetes. A genome-wide association study reported that polymorphisms in Wnt5B, Wnt co-receptors Lrp5/6, and Wnt signalling transcriptional cofactor, TCF7L2, are associated with an increased risk of metabolic syndromes.”

3. Statins are known to cause insulin resistance through other pathways (including reducing prenylation of small GTPases). These mechanisms must be discussed vis a vis the current results. Do the authors ascribe the statin-induced insulin resistance solely to changes in IRS1?

Response: It has been shown that statins induce insulin resistance by inhibiting the insulin signalling pathway, including down-regulation of IRS1 and AKT. Indeed, we observed that

TAZ activates IRS1 and AKT, which are important for GLUT4 membrane localization. In addition, statin-induced IRS1 downregulation was recovered by exogenous active TAZ mutant (TAZ4SA). These results indicate that TAZ is involved in statin-mediated insulin resistance. However, prenylation of small GTPases including Rho and Rab4 facilitates their anchoring to the plasma membrane, which is important for GLUT4 translocation to the plasma membrane for glucose uptake. Statins inhibit HMG-CoA reductase to decrease mevalonate levels and ultimately down-regulate the prenylation of Rho and Rab4 GTPase. Thus, as reviewer indicated, it is also possible that other pathways regulated by statin play a certain role in statin-induced insulin resistance. In this study, we did not investigate the effect of TAZ in Rab4 prenylation, which may yield a robust conclusion regarding the role of TAZ in statin-induced insulin resistance. We included the aforementioned explanation in the discussion section.

Other Issues:

1. On line 92, when referring to figure 2d, the authors state that glucose clearance is attenuated by TAZ KO. However, it appears that the peak glucose concentration in blood was higher in the KO, while the rate of clearance remains the same. Therefore, the rate of clearance does not appear in fig 2d to be altered but rather the initial response and/or the glucose load was higher in the KO. The rate of glucose clearance could be examined using Ki.

Response: Thank you for your suggestion. We revised the description of figure 2d as follows: ‘After glucose infusion, blood glucose disposal decreased in mKO mice’.

2. To ensure that the effects of TAZ are limited to IRS1 expression alone, the authors should examine whether AKT phosphorylation can be rescued by IRS1 overexpression in TAZ KO MEFs or TAZ siRNA treated C2C12.

Response: We rescued IRS1 in TAZ KO MEFs via retroviral transduction of human IRS1-overexpressing vector and assessed AKT phosphorylation upon insulin treatment via immunoblotting (Supplementary Figure 3c and 3d).

Supplementary Figure 3.

3. Fig 5c: the representative image for IRS1 in the knockdown does not match the quantification in Fig 5d. IRS1 is increased in 5c knockdown treated with Wnt3a compared to knockdown untreated but the quantification shows no increase with low variation between repeat experiments.

Response: Figure 5d shows *Irs1* mRNA levels determined via qRT-PCR, not via quantification of immunoblot data. The difference between immunoblot and qRT-PCR data is thought to result from unknown post-translational mechanisms for IRS1 protein stability, induced upon Wnt3a treatment.

4. For APCmKO mice the authors should present control measures of Wnt signalling to show that the KO displays the desired effects of increased Wnt signalling.

Response: We included immunoblot data, which shows increased non-phospho (active) β -catenin level in APCmKO and DbmKO mice. The results verify increased Wnt signalling in the mice because the Wnt signal increases β -catenin stability and facilitates nuclear localization (Figure 5e).

Figure 5.

5. The introduction is scant and lacks biological context.

Response: Thank you for your comment. We included the following description in Introduction section; “Insulin resistance is a condition wherein cells do not respond appropriately to insulin, further characterized by a risk of developing metabolic syndrome such as cardiovascular disease and type 2 diabetes. Skeletal muscles constitute a major organ for insulin-stimulated glucose uptake and disposal under normal conditions. Under physiological conditions, insulin activates glucose uptake by stimulating the canonical IRS-PI3K-Akt pathway, which stimulates glucose transporter (GLUT) 4 translocation to the membrane for glucose uptake.”

6. Fig. 2b shows results of Glut4 in C2C12 myotubes. This is surprising as this cell line is notorious for little or no expression of the transporter even in the myotube stage. Perhaps a validation of the antibody would be useful to strengthen the finding? Second, myoblasts certainly do not express Glut4, and yet glucose uptake was measured in myoblasts (2a). This is confusing and should be done in myotubes.

Response: Per your suggestion, we analysed Glut4 levels in C2C12 myoblasts, myotubes, mouse gastrocnemius muscle, and 3T3-L1 cells. As indicated, we did not observe Glut4 protein expression in myoblasts; however, we observed it in myotubes, muscle tissue, and 3T3-L1 cells. Notably, Glut4 protein levels were significantly increased?? in 3T3-L1 cells, suggesting that our anti-Glut4 antibody is functional. Following is the result for antibody validation (Supplementary Figure 14):

Supplementary Figure 14. Lysates from C2C12 myoblasts, C2C12 myotubes, mouse gastrocnemius muscle, and 3T3-L1 preadipocytes were analysed via an immunoblot assay to detect Glut4. Vinculin was used as a loading control.

In Figure 2a, glucose uptake was assessed in myotubes, not in myoblasts. We described this as ‘C2C12 myoblasts harbouring myotubes’ in the legend of Figure 2a. We apologise for the confusion. We replaced ‘C2C12 myoblast harbouring myotubes’ with ‘C2C12 myotubes’.

Reviewer 1 comments:

In the manuscript "TAZ couples Hippo/Wnt signalling and insulin sensitivity through IRS1 expression", Hwang et al., show that TAZ upregulates IRS1 expression in muscle through interaction with c-Jun. This regulation affects glucose uptake and insulin sensitivity. Moreover, Hwang et al. found that this axis is required for the Wnt induced insulin signalling. Consistent with their results, TAZ inhibition by statins conferred insulin resistance in muscle cells.

The authors provide finely and strong evidence regarding the role of TAZ in the stimulation of insulin sensitivity through up-regulation of IRS1 expression. Moreover, they also clearly demonstrate the requirement of TAZ downstream of Wnt signalling in this context. There are some points that need to be better addressed to support their hypothesis as reported below.

Points:

1- In figure 2C rescue experiment with TAZ overexpression has been performed in MEFs. Since the most important experiments have been done in muscle cells/tissue, the authors should provide similar rescue experiments in C2C12 cells.

2- In figure 3 the authors provide data demonstrating that TAZ engages c-Jun to stimulate Irs1 transcription. Nevertheless, they did not show that c-Jun is required for TAZ induction of Irs1 transcription. Is TAZ inducing Irs1 expression (or Glucose uptake or insulin signalling activation) in cells depleted for c-Jun?

3- Are there evidences about other Hippo pathway components involved in IRS1 regulation/insulin signalling? Please discuss.

4- YAP and TAZ share many functions (frequently they are listed as single YAP/TAZ) it is important to understand what belongs to YAP and what to TAZ. Interestingly the authors in this MS excluded a potential role of YAP based on the experiments in figure 1, in which they monitor total YAP protein levels. Authors must evaluate better this aspect with more and independent evidences.

Responses to Reviewer 1's Comments:

In the manuscript “TAZ couples Hippo/Wnt signalling and insulin sensitivity through IRS1 expression”, Hwang et al., show that TAZ upregulates IRS1 expression in muscle through interaction with c-Jun. This regulation affects glucose uptake and insulin sensitivity. Moreover, Hwang et al. found that this axis is required for the Wnt induced insulin signalling. Consistent with their results, TAZ inhibition by statins conferred insulin resistance in muscle cells. The authors provide finely and strong evidence regarding the role of TAZ in the stimulation of insulin sensitivity through up-regulation of IRS1 expression. Moreover, they also clearly demonstrate the requirement of TAZ downstream of Wnt signalling in this context. There are some points that need to be better addressed to support their hypothesis as reported below.

Points:

- 1) In figure 2C rescue experiment with TAZ overexpression has been performed in MEFs. Since the most important experiments have been done in muscle cells/tissue, the authors should provide similar rescue experiments in C2C12 cells.
- 2) In figure 3 the authors provide data demonstrating that TAZ engages c-Jun to stimulate Irs1 transcription. Nevertheless, they did not show that c-Jun is required for TAZ induction of Irs1 transcription. Is TAZ inducing Irs1 expression (or Glucose uptake or insulin signalling activation) in cells depleted for c-Jun?
- 3) Are there evidences about other Hippo pathway components involved in IRS1 regulation/insulin signalling? Please discuss.
- 4) YAP and TAZ share many functions (frequently they are listed as single YAP/TAZ) it is important to understand what belongs to YAP and what to TAZ. Interestingly the authors in this MS excluded a potential role of YAP based on the experiments in figure 1, in which they monitor total YAP protein levels. Authors must evaluate better this aspect with more and independent evidences.

Response 1: As suggested, the rescue experiments were assessed in C2C12 myotubes. Rescued TAZ in TAZ knockdown C2C12 myotubes restored IRS1 level and its mRNA expression (Supplementary Figure 4a and 4b) and glucose uptake (Supplementary Figure 4c).

Supplementary Figure 4.

Response 2: As suggested, TAZ inducing *Irs1* expression was assessed in c-Jun depleted cells. As shown in Supplementary Figure 8a and 8b, c-Jun depletion significantly decreases IRS1 level and its mRNA expression. The results show that c-Jun plays an important role in TAZ-mediated *Irs1* expression.

Supplementary Figure 8.

Response 3: Thank you for the suggestion. We added followings in discussion; “Recently, multiple members of the Mitogen-activated protein kinase kinase kinase kinase (MAP4K) were identified as parallel Hippo signalling kinases, which phosphorylates LATS kinase. Among them, MAP4K4 has been shown to be a negative regulator of insulin signalling; The

silencing of MAP4K4 in adipocytes elevated the expression of GLUT4. In addition, the silencing in human skeletal muscle prevented tumor necrosis factor- α induced insulin resistance. Interestingly, common polymorphisms of the *MAP4K4* locus are associated with Type 2 diabetes and insulin resistance. These results suggest the important role of Hippo signalling regulators in insulin signalling.”

Response 4: As the reviewer commented, TAZ/YAP shares many functions, though it has been also shown that there are different set of target genes of them. In this study, we have assessed the effect of TAZ alone, not YAP in insulin signalling. As shown in Figure 1, muscle specific TAZ KO mice show decreased IRS1 level without alteration of YAP level.

As the reviewer suggested, we studied *Irs1* expression in YAP knockdown MEF cells. We observed that YAP depletion decreases *Irs1* expression and its protein level (bottom Supplementary Figure 17a and 17b), suggesting that YAP plays a role in insulin signalling. Further study including the regulation of insulin sensitivity in muscle specific YAP KO mice might be an interesting project to address a specific role of YAP in insulin signalling.

Supplementary Figure 17. (a) Depletion of YAP decreased *Irs1* transcription. qRT-PCR was performed in control (Con) and YAP-knockdown (YAPi) MEFs. (n = 3). (b) Knockdown of YAP showed mildly decreased IRS1 level. Cell lysates of Con and YAPi MEFs were assessed by immunoblotting. Vinculin was used as a loading control. For panel a, data are presented as mean \pm SD values. Student’s t-test was performed for statistical significance. * $p < 0.05$; *** $p < 0.005$.

Responses to Reviewer 2's Comments:

1) A more detailed characterization of the Taz-KO mice phenotype should be given. Do muscle specific Taz-KO mice develop any metabolic disorder? From data in supplementary figure 4 it seems that Taz-depletion in muscle fibres does not alter muscle morphology, but those results are based only on immunofluorescence analysis of β -dystroglycan and no quantification of the cross-sectional area is provided. In addition, the authors should analyse if there is a switch of fibre composition (slow vs fast) in KO mice due to impaired glucose uptake. Further analysis of the muscle-specific Taz-null mice at different ages and in response to changes in diet would help to clarify the consequences of impaired glucose uptake in Taz-null skeletal muscle.

2) The authors state that Taz regulates glucose uptake through chromatin binding and transcriptional regulation of Irs1. However, they fall short in presenting any mechanistic evidence showing that down-regulation of Irs1 mediates Taz-null mice deficiency. To this end the authors should perform rescue experiments to show that Irs1 over-expression rescues the Taz-KO phenotype. Moreover, recent work showed that Yap/Taz regulates insulin sensitivity in endometrial cells through post-transcriptional regulation of Irs1 (Wang et al, Am J Cancer Res. 2016). Is this mechanism also conserved in skeletal muscle?

3) The authors performed ChIP-seq analysis using flag-tagged Taz on C2C12 cells. Are those proliferating or differentiated cells? Are they stimulated with insulin? The ChIP-seq data should be further analyzed, including a control input and the raw data should be made available through the relevant databases. Does Taz mainly binds to promoters, gene bodies or intergenic regions? How many Taz targets were identified and why did the author focused on Irs1 only? Do Taz chromatin localization change upon insulin treatment? Previous work by the authors showed that Taz interacts with and activates the muscle regulatory factor MyoD in differentiated muscle cells. Did the ChIP-seq analysis confirmed previous data by showing Taz enrichment at MyoD targets? Finally, ChIP data should be validated also in the mouse model, before and after insulin infusion.

4) A histological analysis of Taz and Irs1 localization in skeletal muscle sections should be performed in basal conditions and upon insulin infusion.

5) In figure 4b-c the authors show co-immunoprecipitation analysis of Taz-Jun interaction in 293T and C2C12 cells. However, in order for this interaction to be relevant in

vivo the authors should perform the Co-IP using skeletal muscle extracts. Similar experiments should be also performed to analyse the presence of Tead4 in the complex, as siRNA-mediated depletion of this factor showed it cooperates with Taz and Jun to regulate Irs1 transcription.

Minor points

- 1) The word “expression” refers to genes, not to proteins. The authors should use the word “levels” when referring to proteins to avoid confusion and to help to identify if they refer to transcriptional or post-transcriptional mechanisms.
- 2) In figure 3a, the “highest reads score” is not a measure of strength of binding. Proper analysis should be performed for the ChIP-seq experiments.
- 3) ChIP analysis in figures 3b and 4a would be better represented as percent of input, instead of relative to IgG condition, to indicate the amount of binding at each condition. A negative control should be also included in those experiments.
- 4) The cross-sectional area in supplementary fig. 4 should be quantified.
- 5) The age of the animals and the muscle analysed should be specified in all experiments.

Response 1: Upon administering standard laboratory chow, we did not observe any metabolic disorder except for a difference in insulin sensitivity. No phenotypic differences were observed in liver and fat tissue. In a high-fat diet, however, we observed several phenotypic differences. TAZ mKO mice showed decreased insulin sensitivity and ability of glucose disposal (Supplementary Figure 6a and 6b). Increased body weight of mKO mice was observed with increased adipocyte size (Supplementary Figure 6c and 6d). Interestingly, mKO mice developed a severe fatty liver phenotype evident from increased fat droplets in liver sections (Supplementary Figure 6e). Thus, these results suggest that TAZ is an important regulator of insulin signalling in muscles, thereby influencing whole-body metabolism.

Supplementary Figure 6

Muscle fibre type composition was analysed on the basis of the expression of marker genes of each fibre type (Type I for slow, Type II for fast), using quantitative real-time PCR. There were no significant differences in muscle fibre type composition between WT and mKO muscle (Supplementary Figure 5d). The cross-sectional area of muscle fibres was measured and a significant difference was not observed between WT and mKO muscle tissues (Supplementary Figure 5b). Muscle weight was also similar in WT and mKO mice (Supplementary Figure 5c)

Supplementary Figure 5

Response 2: As suggested, the rescue experiments were assessed. Rescued IRS1 in TAZ KO MEFs restored AKT activation (Supplementary Figure 3c) and glucose uptake (Supplementary Figure 3d). In our study, TAZ depletion significantly decreased IRS1 levels in muscle tissue, MEF cells, and C2C12 cells. In addition, TAZ overexpression rescued IRS1 levels with a significant induction of *Irs1* expression. Thus, TAZ plays an important role in *Irs1* expression, at least in the muscle tissue, C2C12 myoblasts, and MEFs.

Supplementary Figure 3

As the reviewer suggested, Wang et al. reported that IGF1 increases IRS1 levels, accompanied with phosphorylation of IRS1 at serine 307 and depletion of TAZ and YAP decreases IRS1 levels in endometrial cancer cell lines. We performed a similar experiment in WT and TAZ-depleted C2C12 myoblasts (bottom Supplementary Figure 12). We observed that IRS1 phosphorylation at Ser 307 was increased in control cells (con, scrambled siRNA treated) and TAZ knockdown (Ti, TAZ specific siRNA treated) cells after 5 min of insulin treatment. However, we did not observe a significant increase in IRS1 levels of both cells. Thus, post-transcriptional regulation of IRS1 does not play a major role in insulin-mediated IRS1 induction in C2C12 myoblasts. Notably, the basal level of IRS1 was significantly decreased upon TAZ knockdown cells in the absence of insulin. Thus, based on our other results, *Irs1* expression contributes to basal production of IRS1 in myocytes.

Supplementary Figure 12. Control and TAZ knockdown C2C12 myoblasts were serum-starved for 3 h with serum-free DMEM and 1 nM insulin was administered to the cells. Thereafter, cells were harvested at the indicated time points and analysed via an immunoblot

assay. β -actin was used as a loading control.

Response 3: We used proliferating C2C12 myoblasts for ChIP-seq analysis; these cells were not treated with insulin. The raw data are accessible through the ArrayExpress database (<https://www.ebi.ac.uk/arrayexpress/>). For reviewer access, please use the following login details:

Username: Reviewer_E-MTAB-6764

Password: Ea11vMzY

Upon assessment of peak annotation, we observed that numerous TAZ-binding sites were located in introns or intergenic regions, not in the promoter or exon regions. We could identify 23,804 targets via ChIP-seq analysis (bottom Supplementary Figure 13).

Supplementary Figure 13. Annotation of whole ChIP-sequencing peaks were assessed, and the distribution of annotation is shown as a pie graph (TSS: transcription start site; TTS: transcription termination site).

In this study, we initially investigated whether known insulin signalling components are regulated by TAZ and observed that *Irs1* expression was regulated by TAZ (Figure 1e). Thus, we attempted to identify cis-regulatory elements for the regulation of *Irs1* expression via ChIP-seq analysis. Therefore, the primary purpose of ChIP-seq analysis was the identification of *Irs1* gene regulatory elements at which TAZ binds. Along with *Irs1*, we verified TAZ binding to the regulatory region of known TAZ target gene *Cyr61* and *Myogenin* (bottom

Supplementary Figure 14).

Supplementary Figure 14. ChIP sequencing peaks flanking the well-known TAZ target gene (*Cyr61*) and MyoD target gene (*Myog*) were visualized using the IGV genome browser with reads for histone modification status (H3K27ac, H3K4me3, and H3Kme1). Scale bar = 500 base pairs.

In the mouse model, we observed that TAZ localized at the TBE1 of *Irs1* in the absence of the insulin signal, which was not observed in mKO mice. After 15 min of insulin treatment, a slight increase of TAZ binding at the TBE1 was observed (Supplementary Figure 9). These results suggest that TAZ is important for basal *Irs1* expression.

Supplementary Figure 9.

Response 4: We performed immunohistochemical analysis for TAZ and Irs1 in the skeletal muscle under basal and insulin-stimulated condition. TAZ primarily localized to the nucleus in both conditions and IRS1 was distributed mostly in the cytoplasm. We did not observe a significant difference in TAZ and IRS1 localization between basal and insulin-treated conditions (bottom Supplementary Figure 15).

Supplementary Figure 15. Vehicle- or insulin- (1 U/kg body weight) treated WT mice were euthanised and the gastrocnemius muscle was dissected out, fixed, dehydrated, and embedded in paraffin. Sectioned tissues were analysed via immunohistochemistry with anti-TAZ and anti-IRS1 antibodies to determine the level and localization of each protein. Scale bar = 50µm

Response 5: We obtained co-immunoprecipitation data on mouse skeletal muscle. TAZ and c-Jun or Tead4 interactions were confirmed *in vivo* (Supplementary Figure 10a and 10b).

Supplementary Figure 10.

Response of minor point 1: Thank you for your comments. We revised ‘expression’ to ‘levels’.

Response of minor point 2: We predicted ChIP-seq peaks using Model-based Analysis of ChIP-Seq (MACS) algorithm with a default p-value cut off (10^{-5}). We considered the peak scores derived from the MACS algorithm and histone modification status data available in the GEO database to select major peaks to be analysed further. With these two data sets, TBE1 was selected as the primary target peak for further identification.

Response of minor point 3: Thank you for your comments. We revised the ChIP-qPCR data to represent a percentage of the input and a non-peak negative control was included in Figure 4a.

Figure 4.

Response of minor point 4: As suggested, the cross-sectional area was quantified along with staining data (Supplementary Figure 5b).

Supplementary Figure 5.

Response of minor point 5: As suggested, the age of the all animals used was specified in the figure legends.

Responses to Reviewer 3's Comments:

Hwang et al examine here the role of TAZ/YAP in insulin signalling. Specifically, they found that TAZ is required for basal IRS1 expression and that Wnt-induced insulin sensitivity is mediated through TAZ-induced increase in IRS1 mRNA in skeletal muscle. TAZ was found to act in association with c-Jun and TEAD in order to regulate IRS1 mRNA levels. Second, statin-induced insulin resistance was found to be associated with decreased TAZ and IRS1 expression, while overexpression of TAZ rescued statin-induced insulin resistance.

Strengths:

1. This study establish a new connection between two previously known facts: that Wnt signalling induces IRS1 expression and that Wnt along with nutrient/metabolic sensing pathways regulate TAZ. Thereby, it establishes TAZ as a novel regulator of insulin signalling. The novelty of this work lies in the identification of TAZ-c-Jun and TAZ-TEAD mediated IRS1 expression as well as in the determination of TAZ as a mediator of statin-induced insulin resistance.
2. The experiments performed are clear and methodical in approach, and the findings are convincing. The use of a number of models strengthens the authors' findings, and supportive measures are provided to strengthen each conclusion.

Weaknesses:

1. The study does not provide deep mechanistic insight, it is rather a signalling study which, although very well performed, is not of high cell biology novelty in approach.
2. The physiological context of Wnt regulating IRS1 is not provided.
3. Statins are known to cause insulin resistance through other pathways (including reducing

preylation of small GTPases). These mechanisms must be discussed vis a vis the current results. Do the authors ascribe the statin-induced insulin resistance solely to changes in IRS1?

Other Issues:

1. On line 92, when referring to figure 2d, the authors state that glucose clearance is attenuated by TAZ KO. However, it appears that the peak glucose concentration in blood was higher in the KO, while the rate of clearance remains the same. Therefore, the rate of clearance does not appear in fig 2d to be altered but rather the initial response and/or the glucose load was higher in the KO. The rate of glucose clearance could be examined using Ki.
2. To ensure that the effects of TAZ are limited to IRS1 expression alone, the authors should examine whether AKT phosphorylation can be rescued by IRS1 overexpression in TAZ KO MEFs or TAZ siRNA treated C2C12.
3. Fig 5c: the representative image for IRS1 in the knockdown does not match the quantification in Fig 5d. IRS1 is increased in 5c knockdown treated with Wnt3a compared to knockdown untreated but the quantification shows no increase with low variation between repeat experiments.
4. For APCmKO mice the authors should present control measures of Wnt signalling to show that the KO displays the desired effects of increased Wnt signalling.
5. The introduction is scant and lacks biological context.
6. Fig. 2b shows results of Glut4 in C2C12 myotubes. This is surprising as this cell line is notorious for little or no expression of the transporter even in the myotube stage. Perhaps a validation of the antibody would be useful to strengthen the finding? Second, myoblasts certainly do not express Glut4, and yet glucose uptake was measured in myoblasts (2a). This is confusing and should be done in myotubes.

Response 1: As the reviewer suggested, in our study, we described a novel insulin signalling regulatory pathway involving TAZ and proposed that TAZ links Hippo/Wnt signalling and insulin sensitivity. We observed that TAZ upregulates IRS1 and stimulates Akt and Glut4-mediated glucose uptake in myocytes. In muscle-specific TAZ-knockout (mKO) mice, IRS1 expression and insulin sensitivity were significantly decreased. In addition, the present study reported that statin-derived alterations in insulin sensitivity are mediated by TAZ. Thus, the present results suggest that TAZ downregulation may be a putative cause of statin-driven diabetogenic activity. In addition, we included the phenotype of mKO mice upon administration of a high-fat diet, in the revised manuscript. Interestingly, we observed that mKO mice have increased body weight, adipocyte size, and fatty liver phenotype (Supplementary Figure 6), suggesting that TAZ plays an important role in cellular metabolism via regulation of insulin sensitivity. We hope that these additional data improve the quality of our manuscript.

Supplementary Figure 6.

Response 2: Thank you for your comment. We have included the following contents in the Results section; “Insulin signalling cross-talks with the Wnt signal and altered Wnt signalling components impair glucose metabolism and diabetes. A genome-wide association study reported that polymorphisms in Wnt5B, Wnt co-receptors Lrp5/6, and Wnt signalling transcriptional cofactor, TCF7L2, are associated with an increased risk of metabolic syndromes.”

Response 3: It has been shown that statins induce insulin resistance by inhibiting the insulin signalling pathway, including down-regulation of IRS1 and AKT. Indeed, we observed that TAZ activates IRS1 and AKT, which are important for GLUT4 membrane localization. In addition, statin-induced IRS1 downregulation was recovered by exogenous active TAZ mutant (TAZ4SA). These results indicate that TAZ is involved in statin-mediated insulin resistance. However, prenylation of small GTPases including Rho and Rab4 facilitates their anchoring to the plasma membrane, which is important for GLUT4 translocation to the plasma membrane for glucose uptake. Statins inhibit HMG-CoA reductase to decrease mevalonate levels and ultimately down-regulate the prenylation of Rho and Rab4 GTPase. Thus, as reviewer indicated, it is also possible that other pathways regulated by statin play a certain role in statin-induced insulin resistance. In this study, we did not investigate the effect of TAZ in Rab4 prenylation, which may yield a robust conclusion regarding the role of TAZ in statin-induced insulin resistance. We included the aforementioned explanation in the discussion section.

Response of other issue 1: Thank you for your suggestion. We revised the description of figure 2d as follows: ‘After glucose infusion, blood glucose disposal decreased in mKO mice’.

Response of other issue 2: We rescued IRS1 in TAZ KO MEFs via retroviral transduction of human IRS1-overexpressing vector and assessed AKT phosphorylation upon insulin treatment via immunoblotting (Supplementary Figure 3c and 3d).

Supplementary Figure 3.

Response of other issue 3: Figure 5d shows *Irs1* mRNA levels determined via qRT-PCR, not via quantification of immunoblot data. The difference between immunoblot and qRT-PCR data is thought to result from unknown post-translational mechanisms for IRS1 protein stability, induced upon Wnt3a treatment.

Response of other issue 4: We included immunoblot data, which shows increased non-phospho (active) β -catenin level in APCmKO and DbmKO mice. The results verify increased Wnt signalling in the mice because the Wnt signal increases β -catenin stability and facilitates nuclear localization (Figure 5e).

Figure 5.

Response of other issue 5: Thank you for your comment. We included the following description in Introduction section; “Insulin resistance is a condition wherein cells do not

respond appropriately to insulin, further characterized by a risk of developing metabolic syndrome such as cardiovascular disease and type 2 diabetes. Skeletal muscles constitute a major organ for insulin-stimulated glucose uptake and disposal under normal conditions. Under physiological conditions, insulin activates glucose uptake by stimulating the canonical IRS-PI3K-Akt pathway, which stimulates glucose transporter (GLUT) 4 translocation to the membrane for glucose uptake.”

Response of other issue 6: Per your suggestion, we analysed Glut4 levels in C2C12 myoblasts, myotubes, mouse gastrocnemius muscle, and 3T3-L1 cells. As indicated, we did not observe Glut4 protein expression in myoblasts; however, we observed it in myotubes, muscle tissue, and 3T3-L1 cells. Notably, Glut4 protein levels were significantly increased in 3T3-L1 cells, suggesting that our anti-Glut4 antibody is functional. Following is the result for antibody validation (bottom Supplementary Figure 16):

Supplementary Figure 16. Lysates from C2C12 myoblasts, C2C12 myotubes, mouse gastrocnemius muscle, and 3T3-L1 preadipocytes were analysed via an immunoblot assay to detect Glut4. Vinculin was used as a loading control.

In Figure 2a, glucose uptake was assessed in myotubes, not in myoblasts. We described this as ‘C2C12 myoblasts harbouring myotubes’ in the legend of Figure 2a. We apologise for the confusion. We replaced ‘C2C12 myoblast harbouring myotubes’ with ‘C2C12 myotubes’.

REVIEWERS' COMMENTS:

Reviewer #1 (Remarks to the Author):

The authors addressed the concerns and the MS has been improved, thus is now acceptable in Nat Comm.

Reviewer #2 (Remarks to the Author):

Then authors exhaustively addressed all my concerns. Just as a minor comment, on supplementary figure 5d, the y-axis should be represented as "relative expression" instead of "target ct/reference ct".

I consider this article now suitable for publication in Nature Communications.

Reviewer #3 (Remarks to the Author):

The authors have answered satisfactorily to our comments, including significantly substantiating their model with new experimental evidence.

Responses to Reviewer 2's Comments:

Then authors exhaustively addressed all my concerns. Just as a minor comment, on supplementary figure 5d, the y-axis should be represented as “relative expression” instead of “target ct/reference ct”.

Response: As suggested, the y-axis of Supplementary Figure 5d was represented as “Relative expression”.

Supplementary Figure 5